# Genetic and chemical inhibition of IRF5 suppresses pre-existing mouse lupus-like disease

Tatsuma Ban [1,9 ✉], Masako Kikuchi[1,2,9], Go R. Sato[1,9], Akio Manabe[1], Noriko Tagata[1], Kayo Harita[1], Akira Nishiyama [1], Kenichi Nishimura[2], Ryusuke Yoshimi [3], Yohei Kirino [3], Hideyuki Yanai [4], Yoshiko Matsumoto[5], Shuichi Suzuki[5], Hiroe Hihara[5], Masashi Ito[5], Kappei Tsukahara[5], Kentaro Yoshimatsu[5,8], Tadashi Yamamoto [6], Tadatsugu Taniguchi [4], Hideaki Nakajima[3], Shuichi Ito[2] & Tomohiko Tamura [1,7 ✉]

The transcription factor IRF5 has been implicated as a therapeutic target for the autoimmune disease systemic lupus erythematosus (SLE). However, IRF5 activation status during the disease course and the effects of IRF5 inhibition after disease onset are unclear. Here, we show that SLE patients in both the active and remission phase have aberrant activation of IRF5 and interferon-stimulated genes. Partial inhibition of IRF5 is superior to full inhibition of type I interferon signaling in suppressing disease in a mouse model of SLE, possibly due to the function of IRF5 in oxidative phosphorylation. We further demonstrate that inhibition of IRF5 via conditional *Irf5* deletion and a newly developed small-molecule inhibitor of IRF5 after disease onset suppresses disease progression and is effective for maintenance of remission in mice. These results suggest that IRF5 inhibition might overcome the limitations of current SLE therapies, thus promoting drug discovery research on IRF5 inhibitors.

[1] Department of Immunology, Yokohama City University Graduate School of Medicine, Yokohama, Japan. [2] Department of Pediatrics, Yokohama City University Graduate School of Medicine, Yokohama, Japan. [3] Department of Stem Cell and Immune Regulation, Yokohama City University Graduate School of Medicine, Yokohama, Japan. [4] Department of Inflammology, Social Cooperation Program, Research Center for Advanced Science and Technology, University of Tokyo, Tokyo, Japan. [5] Tsukuba Research Laboratories, Eisai Co., Ltd., Ibaraki, Japan. [6] Cell Signal Unit, Okinawa Institute of Science and Technology Graduate University, Okinawa, Japan. [7] Advanced Medical Research Center, Yokohama City University, Yokohama, Japan. [8] Present address: RIN Institute Inc., Tokyo, Japan. [9] These authors contributed equally: Tatsuma Ban, Masako Kikuchi, Go R. Sato. ✉email: tatban@yokohama-cu.ac.jp; tamurat@yokohama-cu.ac.jp

Systemic lupus erythematosus (SLE) is an intractable chronic autoimmune disease characterized by a breakdown of immune tolerance to nuclear self-antigens[1,2]. The patients are primarily females of reproductive age and manifest the production of autoantibodies, such as those against double-stranded DNA (dsDNA) and ribonucleoproteins. Immune complexes formed by these autoantibodies are deposited in various organs and induce recurrent inflammatory lesions. There are genetic, environmental (infection and sunburn), and hormonal (estrogen) factors that influence the disease development[3].

Current standard therapies for SLE include glucocorticosteroids, antimalarial drugs, and immunosuppressants[4,5]. The most recently approved agent is a monoclonal antibody against B cell-activating factor (BAFF)[6]. Owing to these therapies, the 10-year survival rate has improved from 63 to 91% in the past 50 years[7]. However, SLE patients still have unmet medical needs[8,9]. These problems include the paucity of treatment options for poor responders with intractable organ disease, low health-related quality of life, comorbidities, drug adverse effects such as opportunistic infections and osteoporosis, and a shorter life span. In the case of a relapse, high doses of glucocorticosteroids or immunosuppressants are often required, and therefore relapsing–remitting inflammation results in not only further tissue damage but also severer adverse effects of the drugs.

One of the highly anticipated new drugs currently in clinical trials is an anti-type I interferon (IFN) receptor monoclonal antibody that blocks the action of type I IFNs (IFN-α and -β)[9]. SLE patients display high expression of IFN-stimulated genes (ISGs) called the IFN signature[10], and animal experiments have revealed that blocking of type I IFN signaling inhibits SLE-like symptoms[11–13]. Recently, it was announced that phase III trials of anifrolumab, an antagonist antibody targeting anti-IFN-α and -β receptor subunit 1 (IFNAR1), reached their primary endpoint[14]. The percentage of responders was higher in the anifrolumab group (47.8%) than in the placebo group (31.5%). Nevertheless, the annualized relapse rate was still as high as 43% in the anifrolumab group. Thus, the development of additional therapies other than (or in addition to) blocking of type I IFN signaling is desirable.

The main focus of SLE research has been the adaptive immune system, e.g., aberrant activation of autoreactive T and B cells. Currently, however, it is acknowledged that the innate immune system, which initiates inflammation and actuates the adaptive immune system, also significantly contributes to the disease pathogenesis[15,16]. Indeed, multiple genome-wide association studies have identified *IRF5* as one of the genes whose genetic variants are highly associated with SLE risk[17,18]; *IRF5* encodes a transcription factor called IFN regulatory factor 5, which positively regulates endosomal Toll-like receptor (TLR)-mediated, myeloid differentiation primary response protein 88 (MyD88)-dependent innate immune responses[19,20]. Stimulation of TLR7, TLR8, or TLR9 activates IRF5 via its phosphorylation and nuclear translocation, leading to the induction of type I IFN and inflammatory cytokine genes[21,22]. IRF5 has been shown to be activated in monocytes of most SLE patients, and its expression is further induced by type I IFNs and estrogen[23,24]. Moreover, we and other laboratories have demonstrated that even a half IRF5 deficiency prevents disease onset in various mouse models of SLE[25–28]. IRF5 functions in multiple cell types involved in SLE pathogenesis, for example, conventional dendritic cells (cDCs), plasmacytoid DCs (pDCs), follicular DCs, monocytes, and B cells[17,21]. Therefore, IRF5 appears to be a key factor of various steps of SLE pathogenesis, despite the heterogeneous nature of the disease.

In this study, we address (1) the activation status of IRF5 in the course of human SLE, (2) whether IRF5 functions other than in the induction of type I IFNs, and (3) the effects of IRF5 inhibition on the SLE pathogenesis after the disease onset. We analyze SLE patients and mouse SLE models and develop a prototypical small-molecule compound that limits IRF5 activation. Our results support the idea that IRF5 inhibition overcomes the limitations of current therapies for SLE and should justify drug discovery research on IRF5 inhibitors.

## Results

**IRF5 is activated in both active- and remission-phase SLE (AP-SLE and RP-SLE, respectively) patients.** To find out whether the activation status of IRF5 has relevance to SLE disease activity, we assessed IRF5 nuclear translocation in immune cells isolated from peripheral blood of SLE patients (Fig. 1a, Supplementary Fig. 1a–c, and Supplementary Table 1). Our method, which evaluates the IRF5 nuclear/cytoplasmic ratio, clearly distinguished IRF5 undergoing nuclear translocation (Fig. 1a and Supplementary Fig. 1a–c). Consistent with another report[24], the proportion of cells in which IRF5 was translocated into the nucleus was higher in monocytes from SLE patients than those from healthy donors (Fig. 1b and Supplementary Fig. 2). The proportion of individuals displaying IRF5 nuclear translocation in ≥5% of monocytes was 4% in healthy donors and 52% in SLE patients. Although the difference was modest, a similar tendency was observed in pDCs and cDCs but not B cells (Supplementary Fig. 3a). As expected from other studies[29,30], the degree of IRF5 activation in monocytes correlated with expression levels of ISGs such as *OAS1*, *IFI27*, and *MX1* in peripheral blood (Fig. 1c and Supplementary Fig. 3b).

We then subdivided the patients into two groups according to the SLE disease activity index (SLEDAI) score: AP-SLE (SLEDAI-2K ≥ 5) and RP-SLE (SLEDAI-2K < 5) (Fig. 1d and Supplementary Table 1). The proportion of monocytes featuring IRF5 nuclear translocation and the expression levels of ISGs were higher in AP-SLE than in healthy controls, and furthermore, both remained higher even in RP-SLE than in the controls (Fig. 1e, f and Supplementary Fig. 3c). We also calculated the strength of aberrant IRF5 nuclear translocation by summing the nuclear/cytosolic ratios exceeding the control levels. Still, the degree of IRF5 translocation was largely unchanged between AP-SLE and RP-SLE (Supplementary Fig. 3d). Analysis of publicly available gene expression data from individual SLE patients[31] revealed that ISG expression in peripheral blood was high despite a sign of glucocorticosteroid administration, e.g., glucocorticoid-induced leucine zipper (*GILZ*) expression (Supplementary Fig. 3e). This finding suggested that the persistently high ISG expression after standard therapies is not limited to our study cohort. In RP-SLE, the activation status of IRF5 nuclear translocation in monocytes correlated with serum concentration of residual autoantibodies (Fig. 1g and Supplementary Fig. 3f).

Moreover, regardless of the administration of prednisolone (PSL), hydroxychloroquine (HCQ), or mycophenolate mofetil (MMF), the percentage of monocytes with IRF5 nuclear translocation as well as the expression levels of ISGs remained high (Fig. 1h, i and Supplementary Fig. 3g, h). These results suggested that the current standard therapies are unable to repress aberrant IRF5 activation and ISG expression in SLE patients. Indeed, high doses of representative drugs currently used as standard SLE treatments failed to inhibit IRF5 phosphorylation, another hallmark of its activation, as detected by a phospho-IRF5-specific monoclonal antibody in peripheral blood mononuclear cells (PBMCs) stimulated in vitro with R-848, which is a ligand of TLR7 and TLR8 (Fig. 1j; the validation of antibody specificity is shown in Supplementary Fig. 4a). In these experiments, TPCA-1, an inhibitor of the inhibitor of nuclear

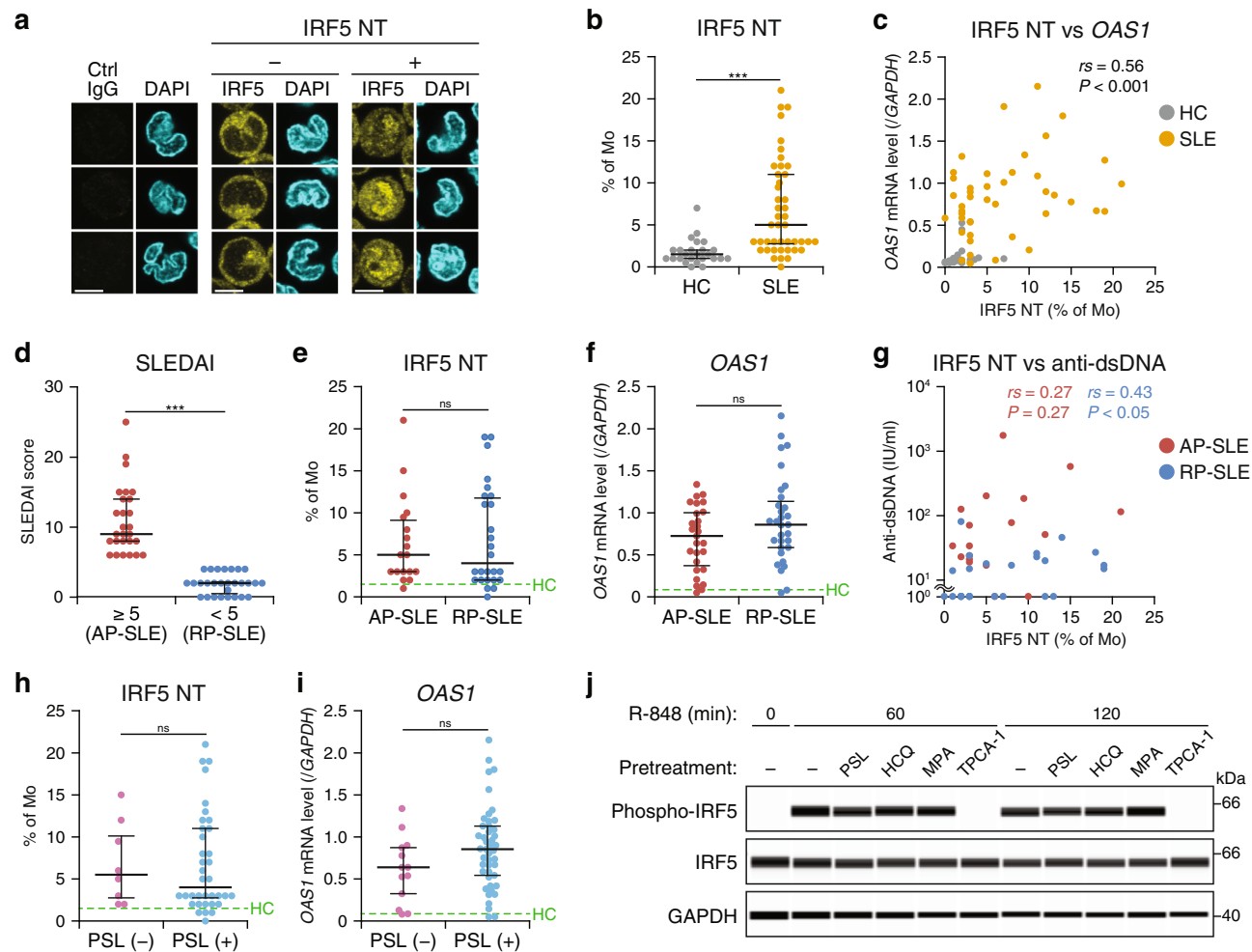

**Fig. 1 Activation status of IRF5 in human SLE. a** Evaluation of IRF5 activation by nuclear translocation (NT). Fluorescent images of monocytes (Mo), stained with control IgG (left) or an anti-IRF5 antibody (middle and right) and then with a fluorochrome-conjugated secondary antibody (yellow), were captured by confocal microscopy. Nuclei were stained with DAPI (cyan). Representative images of cells with or without IRF5 NT, obtained from two of the SLE patients and one of the healthy control (HC) donors, respectively, analyzed in **b** are shown. Scale bars represent 5 μm. **b** IRF5 hyperactivation in SLE-associated monocytes. The IRF5 NT was defined as the nuclear/cytosolic IRF5 ratio of >1.5× most frequent ratio from the same sample containing 75–100 cells (see "Methods"). The proportion of monocytes featuring IRF5 NT in HC donors and SLE patients. **c** Correlation between IRF5 NT and ISG expression. A scatterplot of IRF5 NT prevalence among monocytes and the *OAS1* mRNA level in peripheral blood are shown. **d–f** Persistence of aberrant IRF5 activation and ISG expression in remission-phase (RP)-SLE. Active-phase (AP)- and RP-SLE were defined as the SLEDAI-2K ≥ 5 and <5, respectively (**d**). The percentage of monocytes with IRF5 NT (**e**) and the *OAS1* mRNA level in peripheral blood (**f**) in AP- and RP-SLE were analyzed. **g** Correlation between IRF5 NT and the anti-dsDNA antibody level in RP-SLE. A scatterplot of IRF5 NT prevalence among monocytes and serum anti-dsDNA antibody concentration in AP- and RP-SLE are presented. **h, i** Persistence of aberrant IRF5 activation and ISG expression after the standard therapy. Percentages of monocytes featuring IRF5 NT (**h**) and *OAS1* mRNA levels in peripheral blood (**i**) from SLE patients treated or not treated with PSL were analyzed. **j** Effects of standard-of-care drugs on IRF5 activation. PBMCs from a HC donor were pretreated with 1.5 μM PSL, 3 μM HCQ, 60 μM mycophenolic acid (MPA), or 1 μM TPCA-1 (positive control) for 30 min and then stimulated with 3 μM R-848 for 60 or 120 min. The cell lysates were analyzed by a capillary-based immunoassay with antibodies against phosphorylated IRF5 (phospho-IRF5), total IRF5, and GAPDH as a loading control. Representative data from two independent experiments are depicted. HC: *n* = 25, SLE: *n* = 44 (**b**, **c**), AP-SLE: *n* = 27 (**d**, **f**), or 18 (**e**, **g**), RP-SLE: *n* = 31 (**d**, **f**) or 26 (**e**, **g**), PSL (−): *n* = 8 (**h**) or 13 (**i**), PSL (+): *n* = 36 (**h**) or 45 (**i**). Horizontal bars (**b**, **d–f**, **h**, **i**) represent median with interquartile range. Dashed lines (**e**, **f**, **h**, **i**) indicate the median of HC data. ***$P < 0.001$, ns: not significant (two-sided Mann–Whitney $U$ test). Two-sided Spearman's rank correlation coefficients ($rs$) and $P$ value were used to assess the correlation (**c**, **g**).

factor κ-B (NF-κB) kinase subunit β (IKKβ; the kinase that activates NF-κB and IRF5)[21,22] served as a positive control. TPCA-1 inhibited nuclear translocation of both IRF5 and NF-κB p65 in monocytes and pDCs as expected (Supplementary Fig. 4b, c), indicating that IKKβ inhibitors are not suitable for IRF5-specific suppression.

**Targeting IFNAR1 is not sufficient to suppress disease in a Lyn-deficient mouse SLE model.** The similar dynamics of IRF5 activation and ISG expression in SLE patients prompted us to

investigate whether the inhibition of IRF5 exerts only the action that equals the effect of inhibition of type I IFNs. For this purpose, we used Lyn-deficient ($Lyn^{-/-}$) mice as a mouse SLE model. Lyn deficiency causes IRF5 hyperactivation in the TLR-MyD88 pathway, thereby causing overproduction of cytokines including type I IFNs, and that even a half IRF5 deficiency strongly suppresses the development of SLE-like disease[25]. The production of anti-dsDNA immunoglobulin G (IgG) observed in $Lyn^{-/-}$ mice was almost completely inhibited by either monoallelic or biallelic deletion of *Irf5* ($Lyn^{-/-}Irf5^{+/-}$ or $Lyn^{-/-}Irf5^{-/-}$; Fig. 2a). On the other

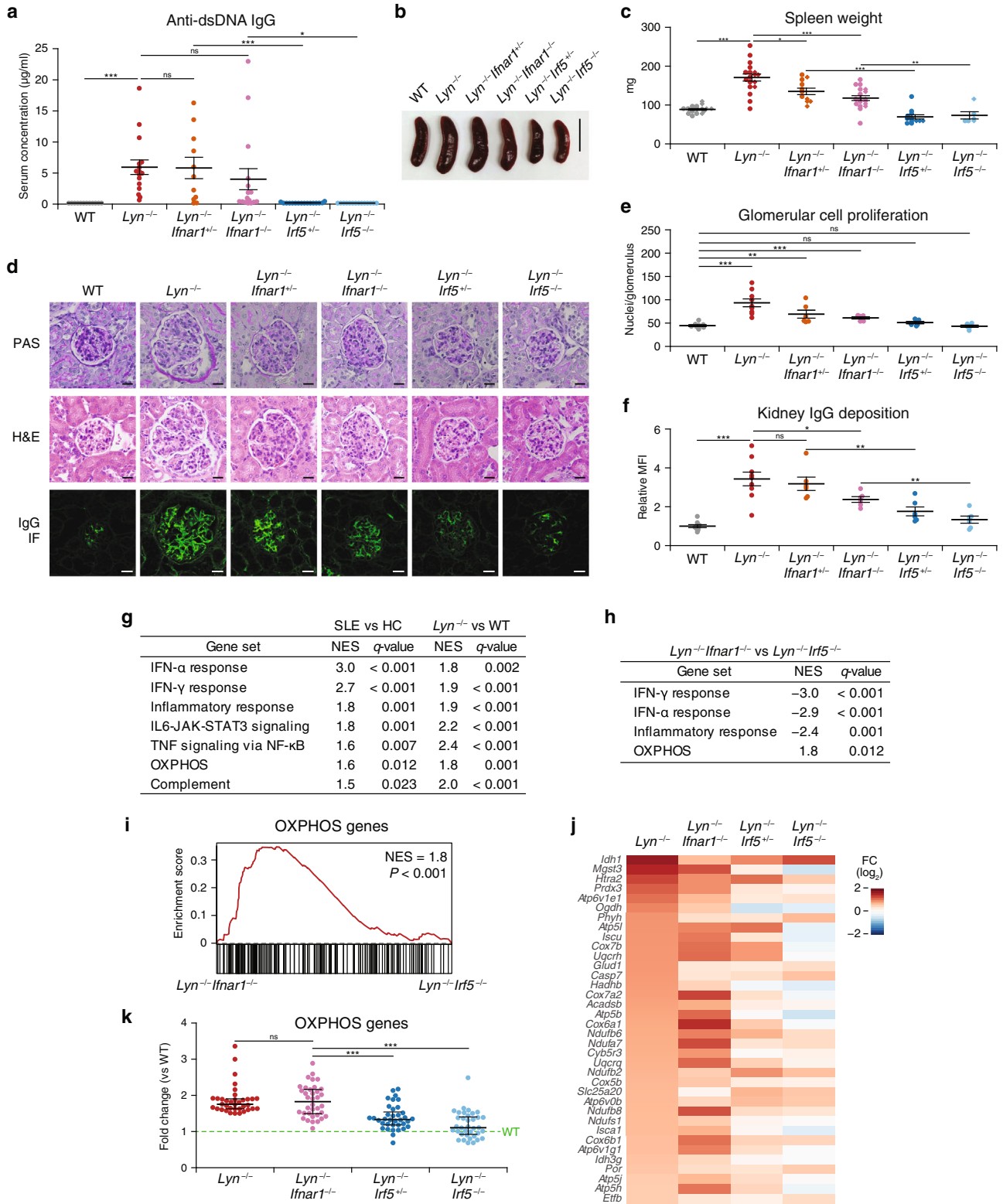

hand, the autoantibody formation was not suppressed either by monoallelic or biallelic deletion of *Ifnar1* (*Lyn*−/−*Ifnar1*+/− or *Lyn*−/−*Ifnar1*−/−; Fig. 2a). Splenomegaly, glomerulonephritis, and IgG deposits were also much more significantly weaker in the *Lyn*−/−*Irf5*+/− strain and *Lyn*−/−*Irf5*−/− strain compared to strains *Lyn*−/−*Ifnar1*+/− and *Lyn*−/−*Ifnar1*−/− (Fig. 2b–f). Similar results were obtained for the activation status of innate and adaptive

immune cells, particularly, the expression of CD80 and CD69 on cDC2s and CD4+ T cells, respectively (Supplementary Figs. 5a, b and 6a, b). Our results are in agreement with the findings obtained in *FcγRIIB*−/−*Yaa* and *FcγRIIB*−/− mouse models of SLE[27]. These results suggested that even partial inhibition of IRF5 is superior to the full inhibition of type I IFN signaling in terms of suppressing the development of an SLE-like disease.

**Fig. 2 Superiority of IRF5 over IFNAR1 in suppressing SLE-like disease development. a** Autoantibody formation. Serum concentration of an anti-dsDNA antibody in WT ($n = 12$), $Lyn^{-/-}$ ($n = 16$), $Lyn^{-/-}Ifnar1^{+/-}$ ($n = 11$), $Lyn^{-/-}Ifnar1^{-/-}$ ($n = 16$), $Lyn^{-/-}Irf5^{+/-}$ ($n = 16$), and $Lyn^{-/-}Irf5^{-/-}$ ($n = 14$) female and male mice at 22–24 weeks of age was measured by ELISA. **b, c** Splenomegaly. Representative image (**b**) and weight data (**c**) of the spleens of WT ($n = 17$), $Lyn^{-/-}$ ($n = 18$), $Lyn^{-/-}Ifnar1^{+/-}$ ($n = 11$), $Lyn^{-/-}Ifnar1^{-/-}$ ($n = 18$), $Lyn^{-/-}Irf5^{+/-}$ ($n = 12$), and $Lyn^{-/-}Irf5^{-/-}$ ($n = 6$) mice at 22-24 weeks of age. Scale bar represents 10 mm (**b**). The circles and diamonds denote the female and male mice, respectively (**c**). **d–f** Kidney pathology and IgG deposition. **d** Representative images of periodic acid-Schiff (top) and hematoxylin–eosin (H&E; middle) staining, and IgG immunofluorescence (IF; bottom) of kidney glomeruli from WT ($n = 9$), $Lyn^{-/-}$ ($n = 9$), $Lyn^{-/-}Ifnar1^{+/-}$ ($n = 6$), $Lyn^{-/-}Ifnar1^{-/-}$ ($n = 6$), $Lyn^{-/-}Irf5^{+/-}$ ($n = 6$), and $Lyn^{-/-}Irf5^{-/-}$ ($n = 6$) mice at 33–34 weeks of age. Scale bars denote 20 μm. **e** Glomerular cell proliferation in **d** (represented by the number of nuclei per glomerulus) was analyzed. Each dot represents the mean nucleus counts of ten glomeruli per mouse. **f** Relative MFI of the IgG deposited in glomeruli in **d** was analyzed. Each dot represents the relative MFI of the five glomeruli per mouse. **g–k** RNA-seq data from peripheral blood obtained from SLE patients and HC donors (public data) or from WT, $Lyn^{-/-}$, $Lyn^{-/-}Ifnar1^{-/-}$, $Lyn^{-/-}Irf5^{+/-}$, and $Lyn^{-/-}Irf5^{-/-}$ female mice at 22–23 weeks of age ($n = 4$ for each genotype) were subjected to GSEA. **g** Sets of genes that were significantly upregulated in both SLE patients (versus [vs] HC donors) and $Lyn^{-/-}$ mice (vs WT mice; false discovery rate < 0.05). **h** Gene sets in **g** were further analyzed to extract gene sets whose expression levels were significantly different between $Lyn^{-/-}Ifnar1^{-/-}$ and $Lyn^{-/-}Irf5^{-/-}$ mice (false discovery rate < 0.05). **i** The GSEA enrichment plot of OXPHOS genes ($Lyn^{-/-}Ifnar1^{-/-}$ vs $Lyn^{-/-}Irf5^{-/-}$). NES normalized enrichment score. **j** A heatmap of the expression of OXPHOS genes in the indicated genotypes. The color represents the mean fold change (FC) as compared to WT. Genes with FC > 1.5 in $Lyn^{-/-}$ mice are shown. **k** A dot plot of the data from **j** ($n = 36$ for each genotype). Horizontal bars indicate mean ± SEM (**a, c, e, f**) or median with interquartile range (**k**). Data in **a–f** were pooled from at least two independent experiments. *$P < 0.05$, **$P < 0.01$, ***$P < 0.001$, ns: not significant (two-sided Student's $t$ test in **a, c, e, f**, and two-sided Mann–Whitney $U$ test in **k**).

**IRF5 regulates oxidative phosphorylation (OXPHOS) in a Lyn-deficient mouse SLE model.** To clarify a role of IRF5 other than the induction of type I IFNs in SLE pathogenesis, we analyzed the peripheral blood transcriptome of SLE patients, which is available in a public database[31], and the peripheral blood transcriptome of Lyn-deficient mice with concomitant *Ifnar1* or *Irf5* deficiency. Gene set enrichment analysis (GSEA) revealed that seven gene sets, including IFN-α response and OXPHOS gene sets, were significantly upregulated both in SLE patients and Lyn-deficient mice compared to healthy donors and wild-type (WT) mice, respectively (Fig. 2g and Supplementary Fig. 6c).

To find the gene sets regulated specifically by IRF5 in the SLE model, we compared $Lyn^{-/-}Ifnar1^{-/-}$ mice with $Lyn^{-/-}Irf5^{-/-}$ mice. We identified only one such gene set, related to OXPHOS, with expression significantly higher in $Lyn^{-/-}Ifnar1^{-/-}$ mice than in $Lyn^{-/-}Irf5^{-/-}$ mice (Fig. 2h, i). This gene set includes genes encoding components of OXPHOS complexes, such as NADH dehydrogenases, cytochrome c oxidase, and reductase, and ATPases (Fig. 2j). The expression of these genes was found to be strongly downregulated in both $Lyn^{-/-}Irf5^{+/-}$ and $Lyn^{-/-}Irf5^{-/-}$ mice (to the WT levels) but were almost unchanged in $Lyn^{-/-}Ifnar1^{-/-}$ mice (Fig. 2k). The elevated expression of IFN-α response genes, most of which are ISGs, was reversed in the $Lyn^{-/-}Ifnar1^{-/-}$, $Lyn^{-/-}Irf5^{+/-}$, and $Lyn^{-/-}Irf5^{-/-}$ strains, with profound suppression in the $Lyn^{-/-}Ifnar1^{-/-}$ strain, which completely lacks type I IFN signaling (Fig. 2h and Supplementary Fig. 6d–f).

To confirm the GSEA results of OXPHOS genes, using tetramethylrhodamine methyl ester (TMRM) as a probe, we measured the mitochondrial membrane potential, which reflects the process of OXPHOS and is therefore a key indicator of mitochondrial activity (Supplementary Fig. 7a, b). Among the various cell types tested, Ly6C$^-$ monocytes and pDCs showed higher TMRM signals in $Lyn^{-/-}$ mice than in WT mice. The upregulation of TMRM signals in these $Lyn^{-/-}$ cell types was significantly suppressed by IRF5 deficiency but not by IFNAR1 deficiency, indicating the role for IRF5. Yet, it is still possible that IRF5 regulates OXPHOS genes in certain subpopulations within other cell types as well. These results suggested that IRF5 is uniquely involved in the induction of OXPHOS genes, causing mitochondrial dysregulation in SLE.

**Conditional *Irf5* ablation reduces disease in a Lyn-deficient mouse SLE model.** The above results, together with previous results obtained from various mouse SLE models[17,25], imply that IRF5 inhibition may serve as an effective therapy for SLE. Nonetheless, these experiments, in which SLE model mice were crossed with IRF5 straight knockout (KO) mice, proved only the prophylactic effects of IRF5 downregulation because IRF5 was deficient before the onset of SLE. To investigate whether IRF5 inhibition is useful for the treatment of the disease after the onset, we first utilized a conditional KO (cKO) of *Irf5* by means of the tamoxifen (TAM)-inducible Cre-loxP (floxed [fl] *Irf5* and *CreER* transgenic) system in Lyn-deficient mice. Serum concentration of anti-dsDNA autoantibodies in *Irf5* cKO ($Lyn^{-/-}Irf5^{fl/fl}CreER$) mice and littermate controls ($Lyn^{-/-}Irf5^{fl/fl}$) was measured every 2 weeks starting from 10 weeks of age. After the anti-dsDNA autoantibody level exceeded 1 μg/ml, the mice were treated with TAM once a day for 5 days by oral gavage (Fig. 3a). Since TAM has been reported to have a certain beneficial effect on SLE regardless of CreER[32], we used $Lyn^{-/-}Irf5^{fl/fl}$ mice treated with TAM as controls. Serum concentrations of anti-dsDNA IgGs before the TAM treatment were comparable between strains $Lyn^{-/-}Irf5^{fl/fl}CreER$ and $Lyn^{-/-}Irf5^{fl/fl}$ (Fig. 3b). We confirmed that the expression of *Irf5* mRNA in peripheral blood was effectively eliminated by TAM treatment of $Lyn^{-/-}Irf5^{fl/fl}CreER$ mice (Fig. 3c). The high expression of ISGs, namely, *Oasl2*, *Ifit1*, and *Isg15*, in $Lyn^{-/-}Irf5^{fl/fl}CreER$ mice was markedly reduced by TAM treatment to the levels comparable to those in WT mice, whereas it persisted in $Lyn^{-/-}Irf5^{fl/fl}$ mice (Fig. 3d and Supplementary Fig. 8a). These results indicated that the conditional deletion of *Irf5* after the appearance of autoantibodies can attenuate the IFN signature in Lyn-deficient mice.

While the anti-dsDNA autoantibody level escalated in $Lyn^{-/-}Irf5^{fl/fl}$ mice over time, this exacerbation was strongly attenuated by TAM treatment in $Lyn^{-/-}Irf5^{fl/fl}CreER$ mice (Fig. 3e). IRF5 has been shown to promote the IgG2c class switch[33,34]. However, total IgG, IgG1, and IgG2c levels in serum samples were mostly comparable between these two strains (Supplementary Fig. 8b), suggesting that the conditional *Irf5* deletion preferentially inhibits the production of autoantibodies. Spleen weight measured at 24–28 weeks after the TAM treatment was significantly lower in $Lyn^{-/-}Irf5^{fl/fl}CreER$ mice than in $Lyn^{-/-}Irf5^{fl/fl}$ mice (Fig. 3f). Similar results were obtained for glomerulonephritis, IgG deposits in kidneys, and the activation status of innate and adaptive immune cells (expression of CD80 and CD69 on cDC2s and CD4$^+$ T cells, respectively) (Fig. 3g–i and Supplementary Fig. 8c, d). On the other hand, the effect of conditional *Irf5* ablation after disease onset in $Lyn^{-/-}$ mice on the

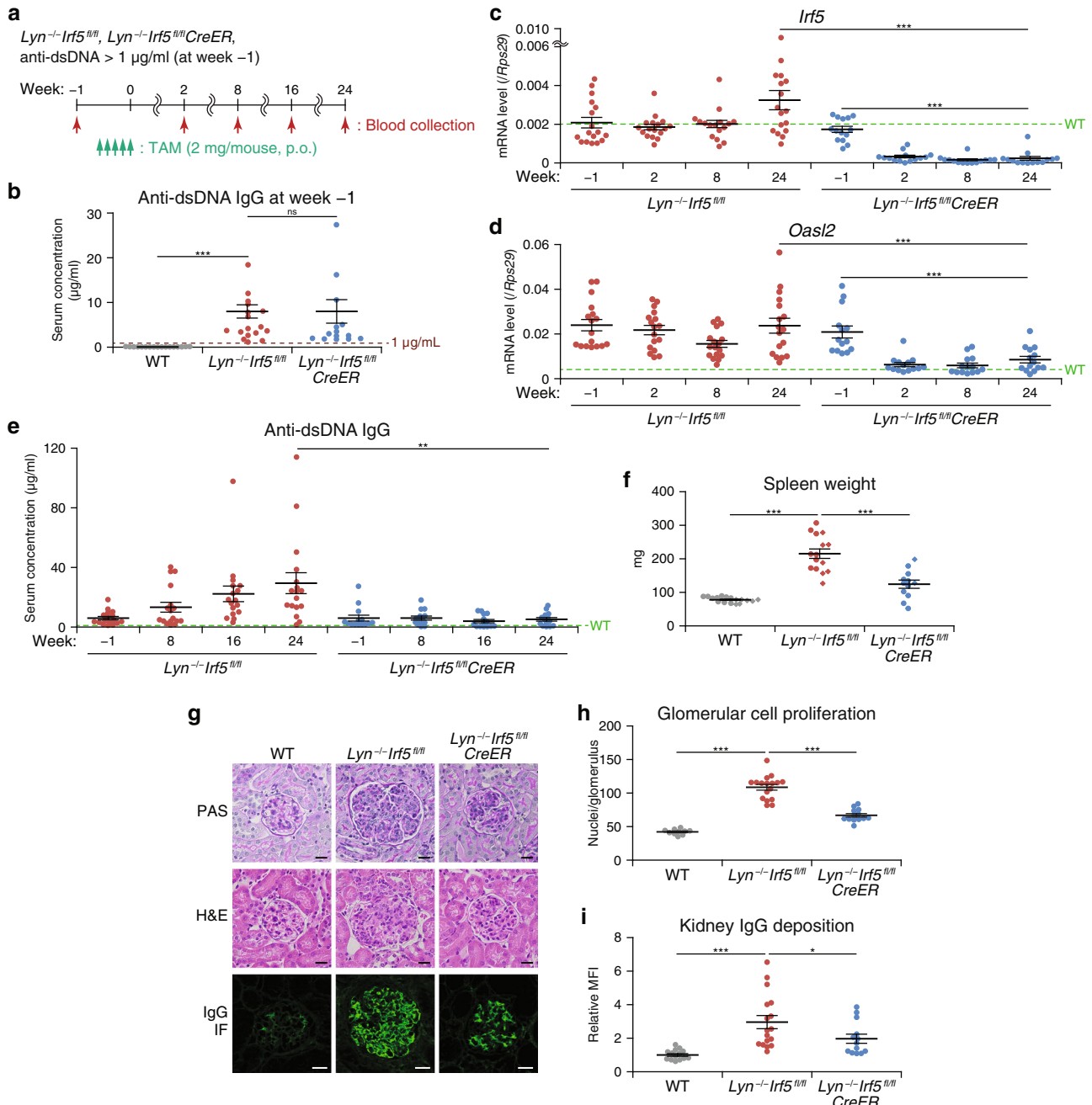

**Fig. 3 Therapeutic effect of *Irf5* ablation on pre-existing mouse SLE-like disease. a** The scheme of the IRF5 cKO experiment; p.o. per os. **b** Concentration of serum anti-dsDNA IgG in WT ($n = 17$), $Lyn^{-/-}Irf5^{fl/fl}$ ($n = 17$), and $Lyn^{-/-}Irf5^{fl/fl}CreER$ ($n = 14$) female and male mice at 1 week before tamoxifen (TAM) treatment (11–23 weeks of age). **c, d** *Irf5* and ISG expression. *Irf5* (**c**) and *Oasl2* (**d**) mRNA levels in peripheral blood from mice in **b** at the indicated time points were analyzed by RT-qPCR. **e** Autoantibody formation. Serum anti-dsDNA IgG concentrations in mice in **b** at the indicated time points were measured by ELISA. **f** Spleen weight of WT ($n = 17$), $Lyn^{-/-}Irf5^{fl/fl}$ ($n = 15$), and $Lyn^{-/-}Irf5^{fl/fl}CreER$ ($n = 12$) mice at 24–28 weeks after TAM treatment (40–51 weeks of age). The circles and diamonds denote the female and male mice, respectively. **g–i** Kidney pathology and IgG deposition analyses as in Fig. 2d–f. WT ($n = 10$ in **h** and 14 in **i**), $Lyn^{-/-}Irf5^{fl/fl}$ ($n = 17$), and $Lyn^{-/-}Irf5^{fl/fl}CreER$ ($n = 13$) mice at age 40–51 weeks were studied. Scale bars represent 20 μm. Horizontal bars (**b–f**, **h**, **i**) indicate mean ± SEM. Dashed lines (**b–e**) denote the mean of WT mouse data. Data were compiled from seven independent experiments. *$P < 0.05$, **$P < 0.01$, ***$P < 0.001$ (two-sided Student's *t* test in **b**, **e**, **f**; two-sided Welch's *t* test in **h**, **i**; two-sided Student's *t* test for a comparison between genotypes or the two-sided paired *t* test for a comparison within the same genotype in **c**, **d**).

aberrant TMRM signals in Ly6C⁻ monocytes and pDCs appeared to be limited (Supplementary Fig. 8e), implying that IRF5 may be involved in a process causing abnormal mitochondrial activity at an early stage of SLE pathogenesis. Overall, these results support the notion that post-disease onset inhibition of IRF5 strongly suppresses the progression of mouse SLE.

**Irf5 cKO maintains the remission of disease in Lyn-deficient mice.** We noticed that the number of long-lived plasma cells (LLPCs), one of the key cell populations responsible for auto-antibody secretion in SLE[35,36], is higher in $Lyn^{-/-}Irf5^{fl/fl}$ mice, and this increase was not effectively attenuated by the conditional deletion of *Irf5* in $Lyn^{-/-}Irf5^{fl/fl}CreER$ mice treated with TAM for

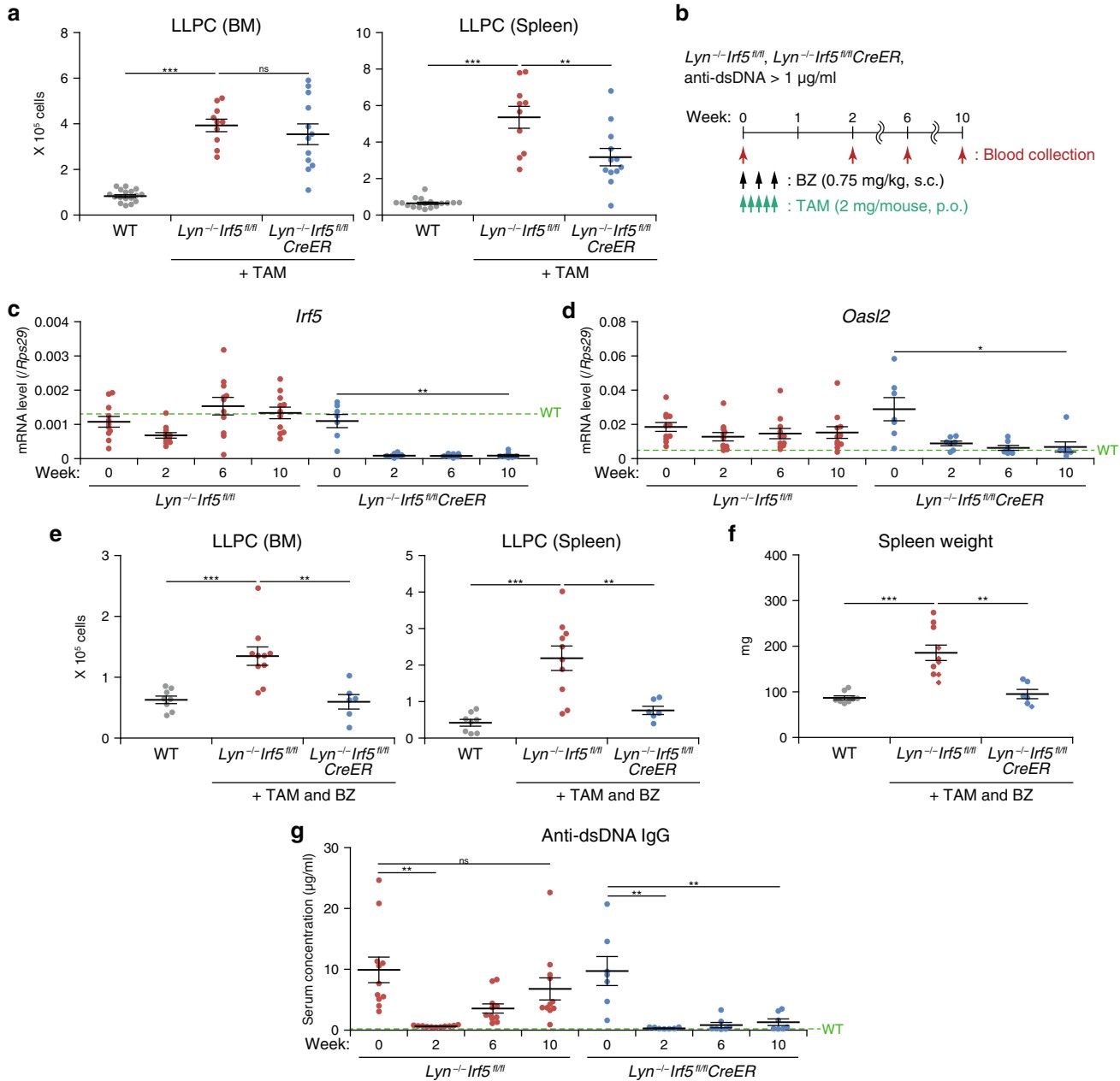

**Fig. 4 Maintenance of remission by the _Irf5_ cKO in the Lyn-deficient mouse model of SLE. a** The LLPC number in bone marrow (BM; left) or in the spleen (right) from WT ($n = 17$), $Lyn^{-/-}Irf5^{fl/fl}$ ($n = 10$), and $Lyn^{-/-}Irf5^{fl/fl}CreER$ ($n = 12$) mice at 24–28 weeks after TAM treatment as in Fig. 3a (40–46 weeks of age) was analyzed by flow cytometry. **b** The scheme of the _Irf5_ cKO combined with BZ-induced LLPC depletion; s.c. subcutaneously. **c**, **d** _Irf5_ and ISG expression. _Irf5_ (**c**) and _Oasl2_ (**d**) mRNA levels in peripheral blood from $Lyn^{-/-}Irf5^{fl/fl}$ ($n = 11$) and $Lyn^{-/-}Irf5^{fl/fl}CreER$ ($n = 7$) female and male mice (29–32 weeks of age at week 0) treated as in **b** were analyzed by RT-qPCR. **e** The LLPC number in BM (left) and spleen (right) from WT ($n = 8$), $Lyn^{-/-}Irf5^{fl/fl}$ ($n = 10$), and $Lyn^{-/-}Irf5^{fl/fl}CreER$ ($n = 6$) mice at 12 weeks after the first injection (41–44 weeks of age) as in **b** was analyzed by flow cytometry. **f** Spleen weight of mice in **e**. The circles and diamonds denote the female and male mice, respectively. **g** Autoantibody production. Serum anti-dsDNA IgG from mice in **c** was quantified by ELISA. Data were pooled from two independent experiments. Horizontal bars (**a**, **c**–**g**) indicate mean ± SEM. Dashed lines (**c**, **d**, **g**) denote mean data from WT mice ($n = 5$). *$P < 0.05$, **$P < 0.01$, ***$P < 0.001$ (two-sided Student's $t$ test in **a**, **e**, **f**; two-sided paired $t$ test in **c**, **d**, **g**).

24–28 weeks (Fig. 4a and Supplementary Fig. 9a). We reasoned that these LLPCs are the source of residual autoantibodies in the TAM-treated $Lyn^{-/-}Irf5^{fl/fl}CreER$ mice. Although it may be difficult to achieve the cure in mouse genetic models of SLE in which the cause of the disease—Lyn deficiency in this case—persists, we explored therapeutic conditions to attain a better effect of IRF5 inhibition. To this end, we utilized bortezomib (BZ), a proteasome

inhibitor that has been shown to deplete LLPCs[37], as a remission induction therapy. A short-term administration of BZ (three times in 5–7 days) to diseased $Lyn^{-/-}$ mice markedly diminished the LLPC population in 1 week (Supplementary Fig. 9b). Serum anti-dsDNA autoantibody levels also dramatically dropped in 2 weeks, reaching a remission-like state (Supplementary Fig. 9c). Nonetheless, 3 weeks after the short-term BZ treatment, the

autoantibody levels began to increase again, suggesting that the remission was transient. Although continued administration of BZ maintained the absence of the autoantibody, the total IgG level declined below the normal level, which could lead to severe adverse effects (Supplementary Fig. 9d).

Based on the above results, we next tested whether the conditional deletion of *Irf5* could restrain the disease flare, which can occur after the transient remission induced by the brief BZ treatment. Accordingly, we administered TAM in combination with BZ to *Lyn*[−/−]*Irf5*[fl/fl] and *Lyn*[−/−]*Irf5*[fl/fl]*CreER* mice (Fig. 4b). The expression of *Irf5* and ISG (*Oasl2* and *Ifit1*) mRNAs in peripheral blood quickly dropped in *Lyn*[−/−]*Irf5*[fl/fl]*CreER* mice (Fig. 4c, d and Supplementary Fig. 9e). Of note, data on *Lyn*[−/−]*Irf5*[fl/fl] mice meant that the induction therapy alone without *Irf5* deletion did not suppress ISG expression; this finding is reminiscent of human SLE patient data in the remission phase (see Fig. 1). Bone marrow and spleen LLPC numbers returned to the high level in *Lyn*[−/−]*Irf5*[fl/fl] mice 12 weeks after the BZ administration, confirming the transient nature of the BZ-induced LLPC depletion (Fig. 4e). On the other hand, when IRF5 was conditionally deleted concomitantly with the induction therapy, i.e., in TAM-treated *Lyn*[−/−]*Irf5*[fl/fl]*CreER* mice, LLPC numbers stayed at the levels comparable to those in WT mice. Similar results were obtained regarding the spleen weight (Fig. 4f). The serum anti-dsDNA autoantibody disappeared 2 weeks after the BZ + TAM treatment in both the *Lyn*[−/−]*Irf5*[fl/fl] strain and *Lyn*[−/−]*Irf5*[fl/fl]*CreER* strain (Fig. 4g). The autoantibody levels, however, returned to the elevated values in the former strain within several weeks, thus mimicking the recurrence. Strikingly, this flare up of autoantibody production was remarkably weaker in the latter strain, in which *Irf5* was conditionally deleted. These results indicated that the IRF5 inhibition indeed effectively maintained the remission state in the mouse SLE model.

**The development of an IRF5 inhibitor**. The therapeutic effects on the Lyn-deficient mouse SLE model obtained by the genetic approach (deletion of *Irf5* after the disease onset) next prompted us to develop IRF5 inhibitors as novel therapeutics for SLE. We performed high-throughput screening (HTS) of approximately 100,000 compounds to identify selective inhibitors of IRF5 transcriptional activity. One small-molecule compound, named YE6144 (its chemical structure is presented in Fig. 5a), which was obtained by analysis of analogs of an HTS hit compound, substantially inhibited the nuclear translocation of IRF5 in monocytes and pDCs treated with R-848 (Fig. 5b). YE6144 inhibited the phosphorylation of IRF5 in both human PBMCs and mouse splenocytes (Fig. 5c). Because IRF5 and NF-κB are known to share their upstream kinase IKKβ in the TLR-MyD88 pathway (also see Fig. 1j and Supplementary Fig. 4b, c for data on the IKKβ inhibitor TPCA-1), we next examined whether YE6144 also inhibits NF-κB activation. Nuclear translocation of NF-κB, however, turned out to be only marginally inhibited in monocytes and was unchanged in pDCs (Fig. 5d). These findings suggested that there exists a mechanism that discriminates between the activation of IRF5 and that of NF-κB in these cell types. A recent study reported that the endolysosomal transporter SLC15A4 and the adaptor protein TASL mediate the selective recruitment and activation of IRF5 by TLR7, TLR8, and TLR9[38]. Although further investigation and structural optimization are necessary to clarify the detailed mechanism of action and to manufacture an agent for clinical use, these results meant that the prototypical IRF5 inhibitor YE6144 selectively suppresses IRF5 activity through inhibition of IRF5 phosphorylation.

We next evaluated the effects of YE6144 on innate immune responses mediated by TLR7 and TLR9, which are known to participate in the SLE pathogenesis via the recognition of nucleic acid-containing immune complexes[39,40]. Induction of type I IFN genes, *Ifnb1* and *Ifna*, in mouse splenocytes stimulated by TLR7 ligands [poly(U) or R-848] or TLR9 ligands (CpG-A or CpG-B oligodeoxynucleotides [ODNs]) was remarkably weakened by the YE6144 pretreatment (Fig. 5e). The production of IFN-β and IFN-α in human PBMCs stimulated with R-848 was decreased by the YE6144 pretreatment in a dose-dependent manner (Fig. 5f). The half-maximal inhibitory concentration ($IC_{50}$) of YE6144 for inhibiting the production of these type I IFNs was approximately 0.09 μM. Essentially the same results were obtained in the analysis of the induction of *Il6*, *Tnf*, and *Il12b* in mouse splenocytes and the production of IL-6 and TNF in human PBMCs (Supplementary Fig. 10a, b).

**An IRF5 inhibitor maintains disease remission in NZB/W F1 mice**. We expected that our IRF5 inhibitor would suppress the progression of mouse SLE, similar to the results obtained following the *Irf5* conditional deletion in Lyn-deficient mice. Indeed, the administration of YE6144 suppressed the exacerbation of autoantibody production in NZB/W F1 mice, another mouse SLE model, both before and after disease onset (Supplementary Fig. 11a–d). Splenomegaly and renal dysfunction were also suppressed by YE6144 treatment after disease onset (Supplementary Fig. 11e–h). Unexpectedly, increases in CD80 on cDC2s, CD69 on CD4+ T cells, and the TMRM signals in Ly6C− monocytes and pDCs in dimethyl sulfoxide (DMSO)-treated NZB/W F1 mice were not evident compared to WT C57BL/6 mice and YE6144-treated NZB/W F1 mice (Supplementary Fig. 11i–k). We infer that the mechanism of the disease in NZB/W F1 mice may be somewhat distinct from the Lyn-deficient mouse model and human SLE.

We further validated whether YE6144 administration maintains the remission in NZB/W F1 mice. BZ was administered two or three times per week to NZB/W F1 mice until the serum anti-dsDNA antibody level dropped below 1 μg/ml to induce remission. After that, we administered YE6144 daily (Fig. 6a). Just as in Lyn-deficient mice, the anti-dsDNA IgG production depleted by BZ injection in NZB/W F1 mice recurred when the vehicle (DMSO) was administered after BZ withdrawal (Fig. 6b). In contrast, the YE6144 administration effectively maintained the remission induced by BZ. Because it has been reported that autoreactive LLPCs constitute a major population in the spleen of NZB/W F1 mice[41], we determined the number of splenic LLPCs 10 weeks after the administration of either DMSO or YE6144. The number of LLPCs returned to the high level in the DMSO group, but this elevation was significantly suppressed by YE6144 (Fig. 6c). Furthermore, other SLE symptoms such as splenomegaly, proteinuria, glomerulonephritis, and aberrant ISG expression, which were seen in DMSO-treated NZB/W F1 mice, were all effectively alleviated by YE6144 treatment of the mice (Fig. 6d–h and Supplementary Fig. 11l, m). These data indicated that IRF5 suppression by the prototypical small-molecule inhibitor YE6144 indeed alleviated the disease course in the NZB/W F1 mouse model of SLE.

Taken together, our results on the two mouse SLE models, Lyn-deficient and NZB/W F1 mice, prove that the post-disease onset inhibition of IRF5 suppresses the disease course and is especially effective in remission maintenance.

## Discussion

In this study, we demonstrated that the excessive IRF5 activation in patients with AP-SLE continues even in those with RP-SLE. The degree of this persistent IRF5 hyperactivation positively correlated with the serum levels of autoantibodies. Consistent

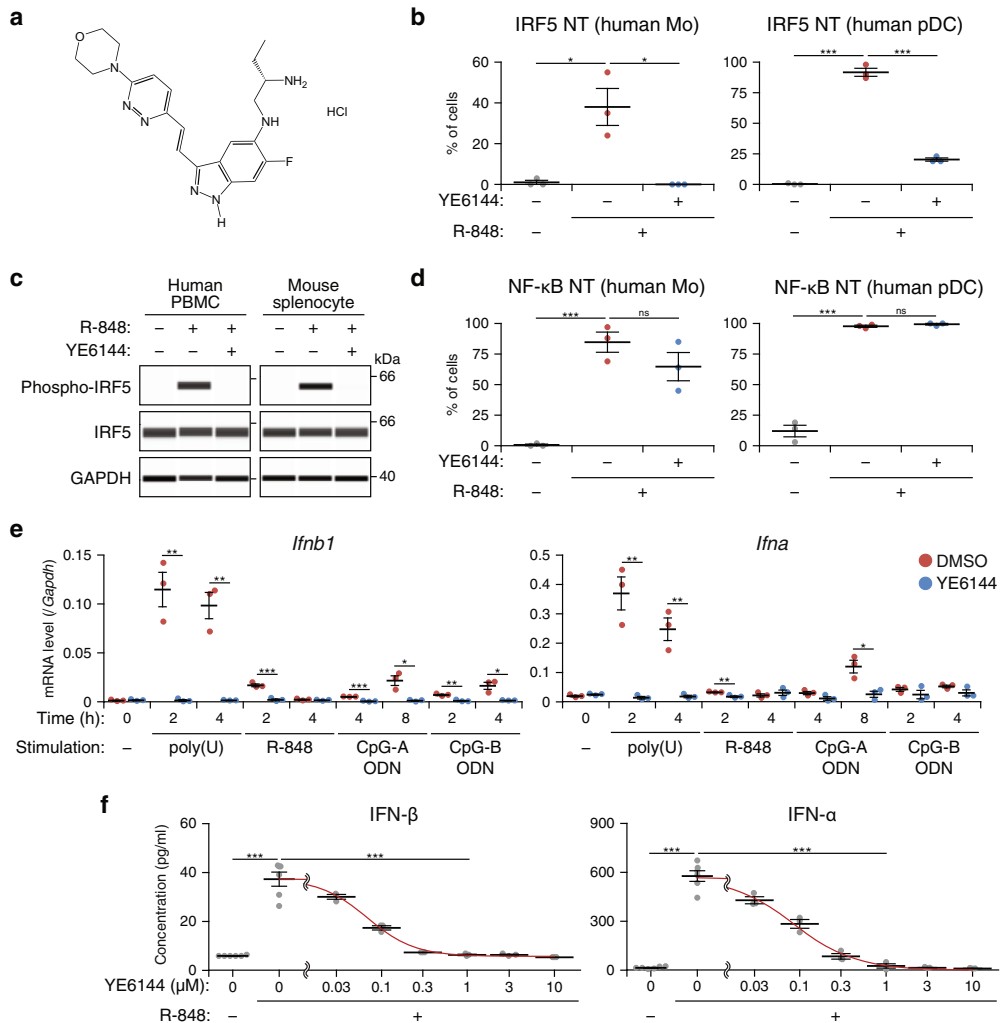

**Fig. 5 Suppression of IRF5 and type I IFN by a prototypical IRF5 inhibitor YE6144. a** Chemical structure of YE6144. **b** IRF5 NT in monocytes (Mo; left) or pDCs (right) sorted from human HC PBMCs that were pretreated with the vehicle (DMSO) or 1 μM YE6144 for 30 min and then were stimulated with 3 μM R-848 for 30 min. **c** IRF5 phosphorylation. Human HC PBMCs and mouse WT splenocytes were pretreated with either 1 and 3 μM YE6144, respectively, or DMSO for 30 min and then stimulated with 3 μM R-848 for 60 min. Cell lysates were analyzed by the capillary-based immunoassay with antibodies against phospho-IRF5, total IRF5, and GAPDH as a loading control. **d** NF-κB p65 NT in cells in **b**. **e**, **f** Type I IFNs. **e** Mouse WT splenocytes were pretreated with either DMSO or YE6144 for 30 min and next were stimulated with 2 μg/ml poly(U), 3 μM R-848, 1 μM CpG-A ODN, or 0.15 μM CpG-B ODN for the indicated period. Total RNA was isolated, and the expression of *Ifnb1* and *Ifna* (detection of 12 subtypes) was analyzed by RT-qPCR. **f** Human HC PBMCs were pretreated with either DMSO (0) or YE6144 at the indicated doses for 30 min and then were stimulated with 3 μM R-848 for 24 h. IFN-β and IFN-α (detection of four subtypes) in culture supernatants was measured by ELISA. The concentration (0.03–10 μM) of YE6144 was plotted on a logarithmic scale. The red line represents a four-parameter log-logistic dose–response curve. Data in **b**, **d** were compiled from three independent experiments (n = 3 in total). Data in **c**, **e**, **f** are representative of two independent experiments (n = 3 in **e** and n = 3 [YE6144] or 6 [DMSO] in **f** for each experiment). Horizontal bars (**b**, **d**–**f**) denote mean ± SEM. *P < 0.05, **P < 0.01, ***P < 0.001 (two-sided Student's t test).

with these findings, IRF5 activation was observed regardless of the presence or absence of standard therapies. These results suggest that the majority of current therapies for SLE are incapable of relieving IRF5 hyperactivation. We can infer that the persistent IRF5 hyperactivation in RP-SLE maintains a state of disease "smoldering." Although the patients with RP-SLE have little or no overt clinical symptoms, IRF5 remains activated in innate immune cells to evoke the production of cytokines, at least a sufficient amount of type I IFNs to induce ISG expression and possibly to promote OXPHOS in mitochondria. These "embers" may lead to the production of residual autoantibodies often detected in the remission phase. Then immune complexes form and again activate IRF5 in innate immune cells, thereby setting up a hidden vicious cycle. Although inflammation per se is restrained

by standard-of-care drugs such as glucocorticosteroids and immunosuppressants, drug discontinuation, infection, or other environmental factors may trigger the amplification of such an IRF5-dependent vicious cycle, causing an SLE flare.

How does the small proportion of monocytes (the first and third quartiles are 2.8 and 11%, respectively, in SLE patients), in which IRF5 is hyperactivated, affect the disease pathogenesis? A previous study has shown that the population of circulating inflammatory $CD5^-CD163^+CD14^+$ cells is expanded in SLE patients and that they produce IFN-α and proinflammatory cytokines when cultured with SLE serum[42]. In the present study, our gating strategy to sort monocytes could not exclude this novel cell type. Therefore, it is tempting to speculate that IRF5 is hyperactivated in these monocytic inflammatory DCs. Another

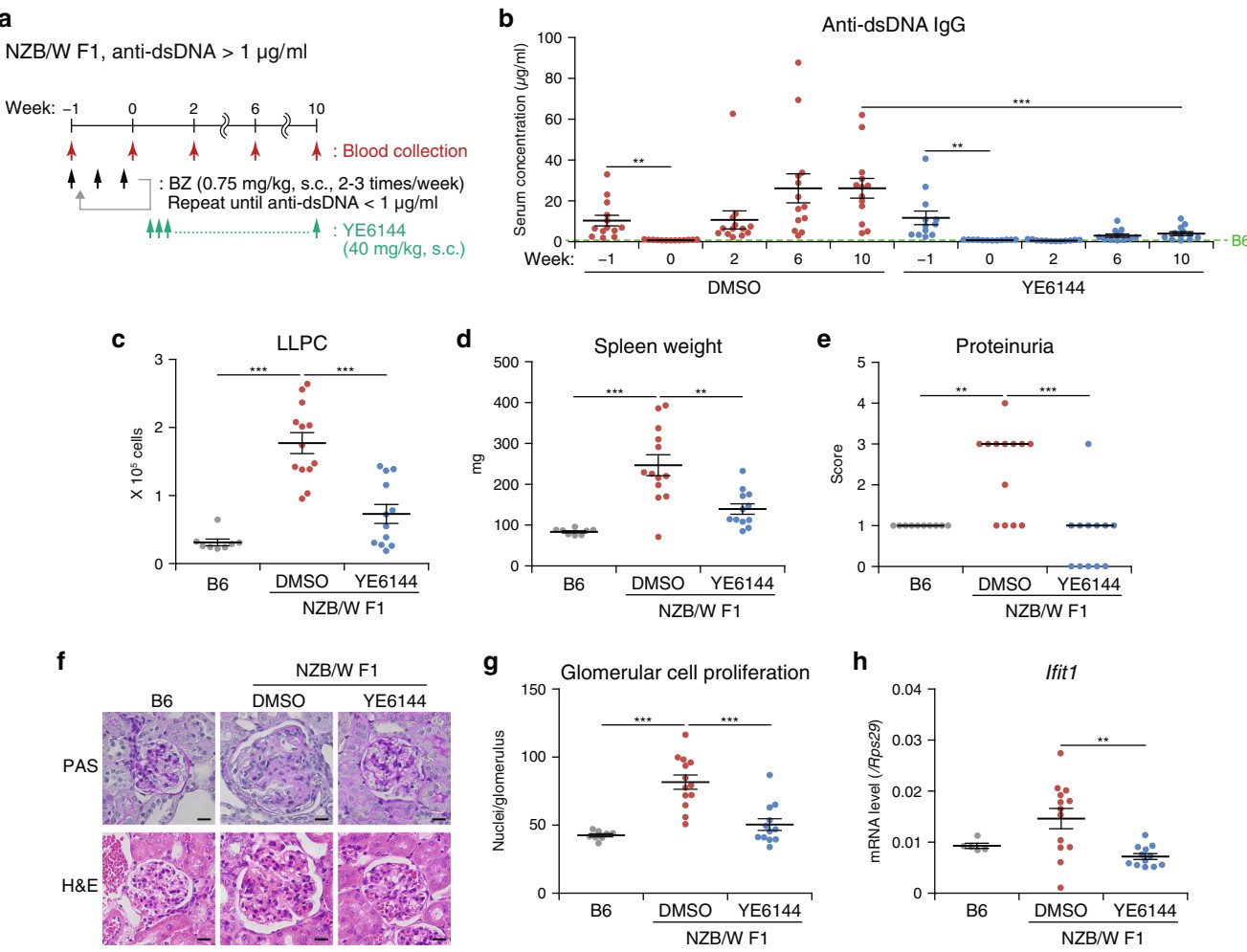

**Fig. 6 Maintenance of remission by YE6144 in the NZB/W F1 mouse model of SLE. a** The scheme of YE6144 treatment combined with BZ administration **b** Autoantibody production. Serum anti-dsDNA IgG levels in NZB/W F1 female mice (31–34 weeks of age at week 0) treated with either DMSO (*n* = 13) or YE6144 (*n* = 12) after BZ injection were analyzed by ELISA. The dashed line denotes the mean data from untreated WT C57BL/6 (B6) mice (*n* = 3). **c** The LLPC number in the spleen from WT B6 mice (*n* = 8) and NZB/W F1 mice in **b** at 10 weeks after the initial YE6144 injection (age 41–44 weeks) was analyzed by flow cytometry. **d** Spleen weight of mice in **c**. **e** The proteinuria score of WT B6 mice (*n* = 10) and NZB/W F1 mice in **c**. **f**, **g** Kidney pathology of mice in **c** analyzed as in Fig. 2d, e. Representative images are shown and scale bars correspond to 20 μm (**f**). **h** ISG expression. The *Ifit1* mRNA level in peripheral blood from WT B6 mice (*n* = 5) and NZB/W F1 mice in **c** was analyzed by RT-qPCR. Data were compiled from three independent experiments. Horizontal bars represent mean ± SEM (**b**–**d**, **g**, **h**) or median (**e**). \*\**P* < 0.01, \*\*\**P* < 0.001 (two-sided paired *t* test in **b** [the same treatment samples]; two-sided Student's *t* test in **b** [different treatment samples], **c**, **d**, **g**, **h**; and two-sided Mann–Whitney *U* test in **e**).

possibility is that most of the immune cells with activated IRF5 migrate to peripheral tissues; hence, the proportion of cells with activated IRF5 appears to be small in peripheral blood.

By means of mouse SLE models, this study addressed the therapeutic implications of IRF5 targeting. IRF5 downregulation was quite effective, whereas IFNAR1 downregulation was insufficient for suppressing the SLE-like disease in Lyn-deficient mice. The recent clinical trial of the IFNAR1 antagonist antibody anifrolumab showed a statistically significant reduction in disease activity versus a placebo in SLE patients[14]. Nevertheless, the reduction was not dramatic, and besides, a considerable rate of relapse was observed. Our results indicate that inhibition of IRF5 may solve this problem. We showed that IRF5 is involved in the induction of not only ISG but also OXPHOS genes and mitochondrial activity in SLE pathogenesis. It has been shown that mitochondrial hyperpolarization induces the production of reactive oxygen species (ROS)[43]. Dysfunctional mitochondria with excessive ROS levels are prone to extrude mitochondrial DNA into the cytoplasm to stimulate the TLR9, cGAS-STING, or

inflammasome pathway[44]. Therefore, we envisage that IRF5-mediated mitochondrial dysfunction might promote the vicious cycle of SLE, in which immune complexes containing autoantibodies and nucleic acids stimulate multiple types of cells and induce cytokines that further activate immune cells[1–3,21]. It will be an interesting future challenge to explore the direct target genes of IRF5 in the relevant cell types, for example, monocytes, pDCs, and the aforementioned CD5−CD163+CD14+ cells.

The conditional *Irf5* deletion in diseased Lyn-deficient mice not only abrogated ISG expression but also completely inhibited the elevation of the serum anti-dsDNA antibody level, indicating that the formation of new autoreactive plasma cells is suppressed, even though the cause of the disease, i.e., Lyn deficiency, persists. It would be worth mentioning that a recent study suggests that autoantibody-secreting plasma cells are generated from CD11c+T-bet+ B cells in a manner dependent on TLR7 signaling[45], which is known to involve IRF5. In fact, IRF5 is reported to be required for the formation of age-associated B cells, a subpopulation of CD11c+T-bet+ B cells[26]. On the other hand, the

conditional *Irf5* deletion was unable to reduce the serum auto-antibody concentration to the level below that observed at the start of *Irf5* deletion, probably owing to the presence of LLPCs. Nevertheless, both the conditional deletion of *Irf5* in Lyn-deficient mice and the administration of the newly identified prototypical IRF5 inhibitor in NZB/W F1 mice revealed that, once LLPCs are depleted by BZ, the IRF5 inhibition can effectively maintain the remission, again, despite the persistence of disease causes. LLPCs are nondividing cells and thus thought to be resistant to glucocorticoids and antiproliferative agents[35,36]. A candidate drug (other than BZ) for the elimination of LLPCs would be atacicept, which inhibits the binding of BAFF and a proliferation-inducing ligand to their receptors including B-cell maturation antigen (BCMA), because plasma cell survival is impaired in BCMA-deficient mice[46]. Although a long-term (52 weeks) clinical trial of atacicept in SLE has been discontinued due to severe adverse effects[47], another clinical trial (short term, 24 weeks) has shown its efficacy without increased risk of severe adverse events[48], suggesting that atacicept is useful for short-term treatment.

Considering our findings in the analyses of SLE patients' samples and mouse SLE models, we envision a novel therapy for SLE using an IRF5 inhibitor as follows. First, IRF5 activation status of PBMCs from SLE patients is assessed as a companion diagnostic method of the IRF5 inhibitor therapy. The patients who display hyperactivation of IRF5 will be referred to the combination treatment with standard remission induction therapies and an IRF5 inhibitor. A long-term remission induced by the IRF5 inhibitor together with standard drugs may eventually result in the disappearance of LLPCs and autoantibodies. If the autoantibody levels remain high, then anti-LLPC drugs such as BZ or atacicept are administered for a short period to induce a deep remission. Eventually, the combination therapy is replaced by IRF5 inhibitor monotherapy for remission maintenance. A recent study reported the efficacy of targeting IRF5 in SLE model mice using a cell-permeable peptide inhibitor[49], which corroborates our current results. In conclusion, our findings suggest that the inhibition of IRF5 may overcome the limitations of current SLE therapies and thereby facilitate drug discovery research on IRF5 inhibitors.

## Methods

**Biological samples from humans.** The research on human donors conducted in this study was approved by the Yokohama City University Certified Institutional Review Board (approval No. A120322001). Informed consent was obtained from all donors or their legal guardians. Peripheral blood from 58 Asian pediatric or adult patients with SLE was obtained at Yokohama City University Hospital. Each patient met at least four of the American College of Rheumatology 1997 revised classification criteria[50]. Peripheral blood from 25 healthy donors was obtained too. Both the SLE patients and healthy donors participated voluntarily in this study. To prevent degradation or modification of the IRF5 protein, protease inhibitor diisopropyl fluorophosphate (Wako) and phosphatase inhibitors cypermethrin (Abcam) and okadaic acid (Santa Cruz Biotechnology) were added into heparinized peripheral blood shortly after blood sampling. Serum was separated from peripheral blood, which was stored at −80 °C until the analysis. PBMCs were isolated using Lymphoprep and SepMate-50 (STEMCELL Technologies) mostly according to the manufacturer's instructions, except that the first centrifugation lasted 20 min and all centrifugation steps were carried out at 4 °C to avoid degradation of IRF5. Total RNA from peripheral blood stabilized in PAXgene Blood RNA Tubes (BD Biosciences) was purified using the PAXgene Blood miRNA Kit (Qiagen, Cat. No. 763134) according to the manufacturers' protocols. Blood test results and clinical data were utilized to calculate disease activity by means of the SLEDAI. AP-SLE and RP-SLE in the patients were defined as the SLEDAI-2K of ≥5 and <5, respectively.

**Mouse biological samples.** All animal experiments were conducted in accordance with Guidelines for the Proper Conduct of Animal Experiments (Science Council of Japan), and all protocols were approved by the institutional review boards of Yokohama City University (protocol Nos. F-A-17-018 and F-A-20-043). *Lyn*−/−, *Irf5*−/−, *Ifnar1*−/−, and *Lyn*−/−*Irf5*+/− and *Lyn*−/−*Irf5*−/− mice on a C57BL/6 background have been described previously[20,25,51,52]. WT (C57BL/6JJcl) and NZB/

W F1 (NZBWF1/Slc) mice were purchased from CLEA Japan and Japan SLC, respectively. *Irf5*fl/fl mice (C57BL/6-*Irf5*tm1Ppr/J; stock No. 017311) and *CreER* mice (B6.Cg-Tg[CAG-cre/Esr1*]5Amc/J; stock No. 004682) were purchased from The Jackson Laboratory. *Lyn*−/−*Ifnar1*+/−, *Lyn*−/−*Ifnar1*−/−, *Lyn*−/−*Irf5*fl/fl, and *Lyn*−/−*Irf5*fl/fl*CreER* mice on a C57BL/6 background were generated through crossing of appropriate strains. Mice were housed in a 12-h/12-h light/dark cycle at 24.0 ± 1.0 °C and a humidity range of 50.0 ± 5.0%. Mouse splenocytes were prepared from a spleen followed by red blood cell (RBC) lysis with ammonium chloride–potassium (ACK) buffer. In case of other than flow cytometric analysis of LLPCs, the spleen was incubated with 1.67 U/ml Liberase DL (Roche) and 0.2 mg/ml DNase I (Roche) for 30 min at 37 °C, prior to RBC lysis. Bone marrow cells were obtained by flushing the femur and tibia, followed by RBC lysis with ACK buffer. Total RNA from peripheral blood stabilized in RNAprotect Animal Blood Tubes (Qiagen) was purified with the RNeasy Protect Animal Blood Kit (Qiagen, Cat. No. 73224) and that from splenocytes using NucleoSpin RNA (Takara, Cat. No. U0955C). Proteinuria was tested by means of Albustix (Siemens). Serum blood urea nitrogen (BUN) levels were quantified via a colorimetric method using a BUN Detection Kit (Arbor assays, Cat. No. K024-H1).

**Reagents.** Mycophenolic acid, the active moiety of MMF, was purchased from Tokyo Chemical Industry. PSL, HCQ, and TPCA-1 were bought from Sigma-Aldrich. Poly(U) (InvivoGen) was complexed with the DOTAP Liposomal Transfection Reagent (Roche) at a ratio of 1 μg of poly(U) per 6 μl of DOTAP and was used for stimulation. R-848 (resiquimod) was acquired from Enzo Life Sciences. CpG-B ODN 1668 (5′-T*C*C*A*T*G*A*C*G*T*T*C*C*T*G*A*T*G*C*T-3′) and CpG-A ODN 2216 (5′-G*G*GGGACGATCGTCG*G*G*G*G*G-3′) were synthesized by Eurofins Genomics (an asterisk indicates a phosphorothioate linkage). The monoclonal antibody that specifically recognizes phosphorylated IRF5 (anti-phospho-IRF5 antibody) was generated by the antibody phage display method[53] following the immunization of rabbits with a synthetic peptide corresponding to the region around phosphorylated serine 446 (S446) of human IRF5 isoform B (equivalent to the S462 of human IRF5 isoform D).

**A quantitative assay of nuclear translocation of transcription factors.** PBMCs ($1–5 \times 10^6$) were stained with fluorescently labeled antibodies to cell surface markers: CD123, CD14, CD141, CD19, CD1c, and CD303 (the fluorochromes, clones, and vendors are listed in Supplementary Table 2). Dead cells were stained with Zombie Red (BioLegend). Cells were then fixed in Cytofix fixation buffer (BD Biosciences) for 10 min at 37 °C and permeabilized with Perm buffer III (BD Biosciences) for 30 min on ice. After washing, the cells were stained with an anti-IRF5 (ab21689; Abcam) or anti-NF-κB p65 (sc372-X; Santa Cruz Biotechnology) rabbit polyclonal antibody. Normal rabbit IgG (Cell Signaling Technology) served for control staining. Then the cells were stained with an Alexa Fluor 488 (AF488)-conjugated anti-rabbit F(ab′)₂ fragment (Jackson ImmunoResearch Laboratories) as a secondary antibody together with a phycoerythrin (PE)-cyanine 7 (Cy7)-conjugated anti-HLA-DR antibody (clone L243, BioLegend). Subsequently, monocytes (CD14+CD19−), B cells (CD14−CD19+), cDCs (CD14−CD19−CD123−HLA-DR+ CD1c+CD141−), and pDCs (CD14−CD19−CD123+HLA-DR+CD303+) were sorted on a FACSAria II instrument (BD Biosciences) with a 100-μm nozzle in 4-way purity mode, using the FACS DIVA software. Nuclei of the sorted cells were stained with 4′,6-diamidino-2-phenylindole (DAPI) Fluoromount-G (Southern Biotech), and next, the cells were transferred into a 384-well glass bottom plate (Greiner) and centrifuged at $400 \times g$ for 5 min. Fluorescent images were obtained on an FV1000-D confocal laser scanning microscope (Olympus) using the FV-10 ASW software. The images were analyzed in Cellprofiler[54]. AF488 signal images were merged with the corresponding DAPI signal images to distinguish nuclear and cytosolic AF488 signals (see Supplementary Fig. 1) to obtain the nuclear/cytosolic mean fluorescence intensity (MFI) ratio. IRF5 was judged to be translocated into the nucleus when the nuclear/cytosolic MFI ratio in the cell was greater than 1.5-fold of the most frequent ratio from the same sample containing 75–100 cells. The total strength of aberrant IRF5 nuclear translocation was calculated by summing the IRF5 nuclear/cytosolic MFI ratios exceeding 1.5-fold of the most frequent ratio. Note that the most frequent IRF5 nuclear/cytosolic MFI ratios between healthy donors and SLE patients were very comparable: 0.58 ± 0.07 and 0.58 ± 0.06 in monocytes, 0.62 ± 0.19 and 0.59 ± 0.12 in pDCs, 0.55 ± 0.10 and 0.56 ± 0.07 in cDCs, and 0.62 ± 0.13 and 0.60 ± 0.11 in B cells, respectively (mean ± SD).

**Reverse transcription quantitative PCR (RT-qPCR).** Total RNA was reverse-transcribed using the PrimeScript RT Reagent Kit (Takara, Cat. No. RR037B) or the PrimeScript II 1st Strand cDNA Synthesis Kit (Takara, Cat. No. 6210 A) according to the manufacturer's instructions. The cDNA was mixed with the Thunderbird SYBR qPCR Mix (Toyobo, Cat. No. QPS-201) and subjected to RT-qPCR on a CFX96 Real-Time System (Bio-Rad) using the CFX Manager software. As per the manufacturer's instructions, a typical two-step PCR protocol was performed (1 min at 95 °C followed by 40 cycles of 15 s at 95 °C and 45 s at 60 °C). Gene-specific primers designed by means of Primer-BLAST (NCBI) are listed in Supplementary Table 3. The data were analyzed by the ΔΔCT method and normalized to glyceraldehyde 3-phosphate dehydrogenase (*GAPDH*), *Gapdh*, or *Rps29* expression levels.

**In vitro treatment of human PBMCs and mouse splenocytes**. Cells were resuspended at $2 \times 10^6$ (human PBMCs) or $4 \times 10^6$ (mouse splenocytes) cells/ml in R10 medium (RPMI 1640 medium [Nacalai Tesque] supplemented with 10% heat-inactivated fetal bovine serum [FBS; Equitech-Bio], 1 mM non-essential amino acids [Thermo Fisher Scientific], 1 mM sodium pyruvate [Thermo Fisher Scientific], 50 μM 2-mercaptoethanol [Sigma-Aldrich], 100 U/ml penicillin [Meiji], and 100 μg/ml streptomycin [Meiji]) and were seeded in tissue culture plates. Thirty minutes after the inoculation, the cells were pretreated with an appropriate reagent and then stimulated with a TLR ligand. The dose and time of each treatment are given in the figure legends.

**Capillary-based immunoassay**. Cells were lysed in RIPA buffer (50 mM Tris-HCl, 150 mM NaCl, 1 mM EDTA, 0.1% sodium dodecyl sulfate, 0.5% of sodium deoxycholate, and 1% NP-40) supplemented with the cOmplete EDTA free protease inhibitor cocktail (Roche) and PhosSTOP phosphatase inhibitor cocktail (Roche). The cell lysates were centrifuged, and the supernatants were subjected to the automated capillary-based immunoassay on a Wes Simple Western System (ProteinSimple) using the Compass software. Anti-IRF5 antibody (clone EPR17067; Abcam), anti-IKKβ antibody (2678 S; CST), and anti-GAPDH antibody (clone 6C5; Abcam) were diluted to concentrations of 25, 1.9, and 10 μg/ml, respectively, with antibody diluent 2 (ProteinSimple). The anti-phospho-IRF5 antibody was diluted to 25 μg/ml with 5% skim milk in Tris-buffered saline containing 0.05% Tween 20. A horseradish peroxidase (HRP)-conjugated anti-rabbit IgG antibody (ProteinSimple) or anti-mouse IgG antibody (Millipore) served as a secondary antibody. Lumino-S/peroxidase (for total IRF5, IKKβ, and GAPDH; ProteinSimple) or Immunostar LD (for phospho-IRF5; Wako) were used as substrates for HRP.

**Plasmids**. The constructs pcDNA3.1-IRF5(isoform B), pcDNA3.1-IRF5(S446A), and pcDNA3.1-IKKβ have been described previously[25]. The cDNA of human IRF5 isoform D (RefSeq accession no. NM_001098629) was obtained by RT-PCR from healthy donor PBMC RNA. The pcDNA3.1-IRF5(isoform D) plasmid was constructed by inserting IRF5 cDNA into the pcDNA3.1(−) vector. Using polyethylenimine (Polysciences), HEK293T cells were co-transfected with empty pcDNA3.1(−), pcDNA3.1-IRF5 (WT isoform B, S446A isoform B, or WT isoform D), and/or pcDNA3.1-IKKβ. Twenty-four hours after transfection, the cells were subjected to a capillary-based immunoassay.

**Enzyme-linked immunosorbent assay (ELISA)**. Anti-dsDNA antibodies and immunoglobulin in mouse serum samples or cytokines in human PBMC culture supernatants were measured by ELISA. For anti-dsDNA IgG analysis, F96 Max-iSorp Nunc-immuno plates (Thermo Fisher Scientific) at 4 °C were coated overnight with 0.05% protamine sulfate (Sigma-Aldrich) and 50 μg/ml calf thymus DNA (Nacalai Tesque) diluted in phosphate-buffered saline (PBS). The plates were washed four times with 0.05% Tween 20 in PBS (PBST) and then incubated with blocking buffer (1% bovine serum albumin [BSA] in PBST) at 20–25 °C for 1 h with shaking. An anti-dsDNA mouse monoclonal IgG2a antibody (clone HYB331-01; Santa Cruz Biotechnology) served as a standard. After the plates were washed with PBST four times, serum samples diluted to 1:1000 with blocking buffer and the standard dsDNA antibody serially diluted with blocking buffer were added and incubated at 20–25 °C for 1 h with shaking. The plates were then washed four times and were incubated for 1 h at 20–25 °C with an HRP-conjugated goat anti-mouse IgG Fc fragment (Bethyl Laboratories) diluted to 1:20,000 with 1% BSA in PBST. After four washes, the tetramethyl benzidine substrate (TMB Substrate Set; BioLegend) was added into wells of the plates, and absorbance at 450/595 nm were recorded on a Sunrise absorbance reader (TECAN). Serum levels of total IgG, IgG1, and IgG2c were determined using ELISA kits purchased from Bethyl Laboratories (Cat. Nos. E90-131, E90-105, and E90-136, respectively). PBMC culture supernatants were analyzed with LumiKine Xpress hIFN-α 2.0 (InvivoGen, Cat. No. luex-hifnav2), Human IFN-beta DuoSet (R&D Systems, Cat. No. DY814-05), Human IL-6 ELISA MAX Standard Sets (BioLegend, Cat. No. 430501), and Human TNF-α ELISA MAX Standard Sets (BioLegend, Cat. No. 430201) ELISA kits.

**Mouse kidney pathology and IgG deposition**. Kidneys freshly excised from mice were either fixed in a neutral buffered 10% formalin solution (Wako) for 2–6 days or snap-frozen in the Tissue-Tek OCT compound (Sakura Finetek Japan) using liquid nitrogen. Formalin-fixed kidneys were embedded in paraffin by means of a closed self-locking embedding device, Tissue-Tek VIP5 Jr. (Sakura Finetek Japan). Freshly frozen blocks were stored at −80 °C until cryosection preparation. Paraffin-embedded kidneys were sectioned and stained with hematoxylin and eosin (H&E) or periodic acid-Schiff reagents by Tokushima Molecular Pathology Institute. The images of stained kidney sections were obtained using Biozero BZ-8000 (Keyence) and BZ Viewer software. Glomerular cellularity was assessed by calculating the mean number of nuclei in ten glomeruli from the H&E-stained cross-sections per mouse using the ImageJ software (NIH)[55]. IgG deposition in glomeruli was assessed by immunofluorescent staining of 4 μm kidney cryosections prepared on a cryostat. Cryosections mounted on Superfrost slide glass (Matsunami) were immediately fixed in −80 °C acetone for 30 s, rinsed in PBS, and blocked with PBS

containing 5% skim milk (Morinaga) at 20–25 °C for 30 min. The sections were then incubated with biotinylated anti-mouse IgG (Vector Laboratories) in blocking buffer at 4 °C overnight. After that, the sections were washed three times with PBS and stained with AF488-conjugated streptavidin (Life Technologies) and then with Hoechst 33342 (Life Technologies). After a rinse with PBS, the sections were mounted with the Prolong Gold Antifade Reagent (Invitrogen). Fluorescent images were scanned by means of FV1000-D. The MFI of each glomerulus was calculated using the ImageJ software and the MFI of five glomeruli were averaged. The relative MFI was determined as the average MFI normalized to the average MFI in the WT samples.

**RNA sequencing and bioinformatics analysis**. Total RNA from peripheral blood was prepared for five mice of each of the following mouse strains: WT, $Lyn^{-/-}$, $Lyn^{-/-}Ifnar1^{-/-}$, $Lyn^{-/-}Irf5^{+/-}$, and $Lyn^{-/-}Irf5^{-/-}$. The RNA samples were quantified on a nanophotometer (Implen), and their quality was assessed using the Agilent RNA6000 Pico Kit (Cat. No. 5067-1513) and Bioanalyzer in accordance with the manufacturers' instructions. All samples had an RNA integrity number >7.2. Libraries were prepared from 500 ng of total RNA by means of SureSelect Strand-Specific RNA Library Prep for Illumina Multiplexed Sequencing (Agilent Technologies). The size of final cDNA libraries was validated using the Agilent High Sensitivity DNA Kit (Cat. No. 5067-4626) and Bioanalyzer in line with the manufacturer's instructions. Concentrations of individual libraries were validated by qPCR with the KAPA Library Quantification Kit (Nippon Genetics, Cat. No. KK4824). Pooled libraries were subjected to the sequencing using the Illumina NextSeq 500/550 High Output Kit v2 (Cat. No. 20024906) and NextSeq 500 System. Illumina Real-Time Analysis software was used for base calling. All sequenced reads were converted to FASTQ format in the bcl2fastq software, followed by mapping to the mouse genome reference sequence mm10 using STAR[56]. Raw read counts were obtained in featureCounts[57]. After normalization to the trimmed mean of $M$ values by the calcNormFactors function in edgeR[58], final read counts were obtained by the rpkm function. GSEA was performed as described elsewhere[59] using the hallmark gene sets[60].

**In vivo treatments of mice**. TAM (Sigma-Aldrich) was dissolved at 100 mg/ml in 99.5% ethanol, and the aliquots were stocked at −20 °C. Prior to administration, the aliquots were further diluted with corn oil (Sigma-Aldrich) and then administered to mice by oral gavage (2 mg/250 μl/mouse, once a day for 5 days). BZ (Wako) was dissolved at 20 mg/ml in DMSO, stocked at −20 °C, diluted with PBS, and injected subcutaneously at a dose of 0.75 mg/(kg body weight) in accordance with the treatment scheme described in the figures. YE6144 was dissolved in PBS containing 10% DMSO and subcutaneously administered daily to NZB/W F1 mice at a dose of 40.0 mg/(kg body weight).

**Flow cytometry**. The mouse splenocytes and bone marrow cells were preincubated with anti-mouse CD16/CD32 (BioLegend) antibody to block Fcγ receptors and then stained with the antibodies (Supplementary Table 2) in 2% FBS at 4 °C for 30 min. To measure the mitochondrial membrane potential, stained cells were incubated with 50 nM TMRM (Thermo Fisher Scientific) in R10 medium at 37 °C for 30 min. The TMRM fluorescence was detected using the PE channel. Dead cells were stained with 7-aminoactinomycin D for 5 min. Data were acquired on BD FACSCanto II (BD Biosciences) using the FACS DIVA software and analyzed in the FlowJo software (Tree Star).

**Screening of IRF5 inhibitors**. A reporter system that distinguishes the transcriptional activities of IRF5 and NF-κB was established and used for HTS. A library containing ~100,000 compounds was subjected to the screening via the reporter system to identify the compounds that selectively inhibit IRF5 transcriptional activity, thereby yielding 26 small-molecule hit compounds. Ten of these hit compounds were validated by endogenous gene expression analysis. One of the validated hit compounds strongly inhibited IRF5 activation at a dose much different from its toxic dose. After that, several analogs of the confirmed hit compound were assessed. One of these compounds, YE6144, was found to have better photoisomerization properties and less toxicity in vivo.

**Statistical analysis**. All statistical analyses were performed in the R software. For a comparison of non-normally distributed data, Mann–Whitney $U$ test was carried out. For a comparison of normally distributed data, either Student's $t$ test or Welch's $t$ test was conducted. For a paired comparison of data, paired $t$ test was performed. For correlation analysis, Spearman's rank-ordered correlation was utilized. Data with $P$ values < 0.05 were considered statistically significant.

**Reporting summary**. Further information on research design is available in the Nature Research Reporting Summary linked to this article.

## Data availability

The RNA sequencing (RNA-seq) data have been deposited in the Gene Expression Omnibus (GEO) database with the primary accession code GSE146334. The public RNA-

seq data from SLE patients and healthy donors[31] were retrieved from the GEO database with the primary accession code GSE72509. Source data are provided with this paper.

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

## Acknowledgements

We thank Dr. Masatoshi Nakazawa, Mr. Masahiro Yoshinari, and Mr. Reiya Nagata at Yokohama City University and Dr. Miho Ohsugi at the University of Tokyo for their help with the experiments. This research was supported by Japan Agency for Medical Research and Development (AMED) under the grant number JP19ek0410042 and JP20ek0410081 (to T. Tamura); the Fund for Creation of Innovation Centers for Advanced Interdisciplinary Research Areas Program in the Project for Developing Innovation Systems from the Ministry of Education, Culture, Sports, Science and Technology (MEXT)/Japan Science and Technology Agency (a grant to T. Tamura), which accompanies matching funds from Eisai Co., Ltd.; Grants-in-Aid (KAKENHI) from the MEXT/Japan Society for the Promotion of Science (grant number 16K19161; to T.B.); a Grant for Strategic Research Promotion from Yokohama City University to T. Tamura; and a grant from Yokohama Foundation for Advancement of Medical Science to T.B.

## Author contributions

T.B., M.K., and T. Tamura designed the study; T.B., M.K., G.R.S., A.M., N.T., K.H., Y.M., and S.S. conducted the experiments; T.B., M.K., G.R.S., A.M., N.T., K.H., A.N., K.N., R.Y., Y.K., H.Y., Y.M., S.S., H.H., M.I., K.T., K.Y., T.Y., T. Taniguchi, H.N., S.I., and T. Tamura analyzed the data; T.B., M.K., and T. Tamura wrote the manuscript; K.N., R.Y., Y.K., H.Y., Y.M., S.S., H.H., K.T., K.Y., T.Y., T. Taniguchi, H.N., and S.I. provided key materials; K.N., R.Y., Y.K., H.Y., Y.M., S.S., H.H., K.T., K.Y., T.Y., T. Taniguchi, H.N., and S.I. provided critical suggestions for the study; and T. Tamura supervised the project.

## Competing interests

Y.M., S.S., H.H., M.I., and K.T. are employees of Eisai Co., Ltd. T. Tamura received joint research funds from Eisai Co., Ltd. All other authors declare no competing interests.
