## [Peer Review File · Nature Communications]

REVIEWER COMMENTS

Reviewer #1 (Remarks to the Author):

Immunosuppressive agents widely used for treating SLE globally inhibit the immune system which predispose patients to severe infection and morbidity. Although significant progress made towards developing targeted therapies for SLE, and several novel therapeutics have reached Phase 2 and 3 clinical trials, to date, only one drug with modest efficacy (Belimumab) has been approved by the FDA. Much anticipated type I IFN blocking therapy has proven some success, but not satisfactory, thus necessitating the development of an improved therapy for SLE. A major roadblock to the development of therapeutics that are specific, effective and have fewer or no side effects is our poor understanding of the cell-intrinsic mechanisms of innate and adaptive immune cells that promote SLE pathogenesis. Delineation of novel critical signaling mechanisms involved in SLE is required to identify therapeutically targetable factors for SLE treatment. With that in mind the current manuscript by Ban et al. determined the potential for targeting IRF5 as an improved therapeutic target for SLE. Studies carried out by this group using both SLE patient samples and mouse models of SLE indicated IRF5 to be a superior therapeutic target for SLE than blocking type I IFN signaling. The authors demonstrated that blocking *Irf5* could be an effective therapeutic target during the onset or the remission phase of disease.

Major Comments:

(1) Although the authors have convincingly demonstrated the potential for IRF5 to be a better therapeutic target for SLE, the cell-type specific mechanisms by which IRF5 and type I IFN signaling may differentially regulate innate and adaptive immune responses to promote SLE autoimmunity was not investigated. The discovery that IRF5 regulates pathways other than just driving type I IFN response is significant and novel, and therefore, delineating mechanisms differentially regulated by IRF5 and type I IFNs (IFN-I) in a cell-type specific manner will fill gap in our knowledge of IRF5 and IFN-I functions.

(2) Through the analysis of transcriptomes, the authors have identified OXPHOS metabolic genes to be differentially regulated in peripheral blood cells between healthy control and SLE patients, and *Lyn*^{-/-}*Ifnar1*^{-/-} and *Lyn*^{-/-}*Irf5*^{-/-} mice. However, no experiments were performed to validate these genes at the protein levels in a cell-type specific manner.

(3) In Figure 3, the authors used tamoxifen inducible *Irf5* conditional knockout system to demonstrate at what stage of targeting IRF5 may have therapeutic benefit. Although the authors focused on autoantibody and type I IFN responses, and kidney pathology, they have missed opportunities to determine the innate and adaptive cellular immune responses to see what cell populations might be targeted.

(4) They have identified a novel prototypical inhibitor for IRF5 which can inhibit IRF5 activation, type I IFN induction and ameliorate kidney disease. Again, no cellular responses were investigated to determine the cell type specific effects of this inhibitor.

(5) It is not clear whether the novel IRF5 inhibitor the authors have identified ameliorates disease by dampening the OXPHOS metabolic activity in B cells or myeloid cells (such as monocytes or DCs)

In conclusion, although authors have identified IRF5 to be a superior therapeutic target for SLE than type I IFN blocking therapy, the manuscript lacks cell-intrinsic mechanisms through which IRF5 targeting may have better therapeutic effects than type I IFN blocking approach. Understanding the differential cell intrinsic mechanisms of IRF5 and type I IFN signaling will fill gap in our knowledge about the role of IRF5 beyond just driving type I IFN response.

Reviewer #2 (Remarks to the Author):

This paper expands previous knowledge by presenting a prototypical small-molecule compound that limits IRF5 activation to treat lupus in mice and hopefully in humans. In addition they report that the activation of IRF5 occurs in people with active and inactive disease and treatment does not matter. The efficacy of the IRF5 depends on the presence of a proteasome inhibitor and this should be stressed in the abstract.

1. Fig. 2, Does the monoallelic *Irf5* mouse have any *Irf5*? If so how much. How much should be

present to reverse the beneficial effect?

2. The monoallelic IRF5 animals should be included in Fig 2g through 2j.

3. Figure 3, would KO of IRF5 in B cells accomplish the same effect? The Lyn^{-/-} mouse develops lupus on a mechanism that starts directly from B cells (removal of the inhibitory effect of the kinase Lyn) and it is probable that IRF5 has a major effect on B cells at least in this model.

4. In Fig 5, dose curves for the drug should be provided.

5. Fig 6. What is the effect of the IRF5 inhibitor alone (before and after the initiation of the disease).

6. Would a proteasome inhibitor enhance the effect of IFN blockade? If that would be true then the authors can claim a better effect for their IRF5 inhibitor.

Reviewer #3 (Remarks to the Author):

This manuscript clearly shows a pivotal role of persistent IRF5 hyperactivation in the initiation and perpetuation of SLE manifestations by in vitro studies of SLE patients and by using two lupus models of Lyn deficient mice and NZBxNZW F1 mice. Results are striking that partial inhibition of IRF5 exhibited superior efficacy than full inhibition of type I interferon signaling in suppressing murine lupus development, which has important clinical implications and highlights IRF5 as an excellent therapeutic target for a number of autoimmune diseases with a strong component of type I interferon mediated disease manifestations. The authors further developed a prototype small-molecule inhibitor of IRF5 activation and showed specific inhibition of nuclear transport and activation of IRF5. Treating established disease with either the inhibitor or conditional ablation of IRF5 could result in sustained remission of both murine models of lupus, which is impressive data to this reviewer.

The strength of the manuscript is its innovation and clinical significance that has direct implications to disease management of patients affected with SLE. The authors offered a mechanistic explanation on consequences of IRF5 hyperactivation in peripheral blood cells relating to expression levels of OXPHOS genes; however, it is not clear how this new information fits into the overall scheme of lupus pathogenesis at present. An important reagent used in this study is a monoclonal antibody to phosphorylated IRF5 that appeared to be made by the authors who immunized rabbits with a peptide containing phosphorylated S446 of human IRF5 isoform B. It would be important for the authors to demonstrate specificity of this reagent in the supplemental materials, and to clarify how this S446 residue relates to the Serine 462 residue commonly described in the literature that its phosphorylation results in IRF5 dimerization and nuclear translocation. Major findings of this manuscript corroborates with two recent publications that one links endosomal TLRs to the recruitment and activation of IRF5 independent of NF- κ B or MAPK via SLC15A4 and TASL (PMID:32433612) and the other on the development of cell-penetrating peptide to inhibit IRF5 activation and function in disease (PMID: 32440537), and the inclusion of these two could strengthen the Discussion of this manuscript. Additional specific comments are as follows:

1. The sex of the mice used should be stated. The authors showed spleen weights of the mice, which might be affected if the mice had different body weights, especially if both sexes were used. A common way to eliminate potential confounding is to present the spleen weight to body weight ratios.

2. The authors measured proteinuria using Albustix, which is not very accurate. If urine samples are available, I would suggest to measure albumin creatinine ratio. Alternative, levels of blood urine nitrogen could be assessed in the serum samples.

Reviewer #4 (Remarks to the Author):

The manuscript by Ban et al. investigates the activation status and function of the transcription factor IRF-5 during systemic lupus erythematosus (SLE) in SLE patients and in a murine model of SLE. Analysis of IRF-5 and interferon-stimulated gene expression revealed that both are activated not only in active SLE patients but also in patients in remission. Using a murine model for SLE (Lyn^{-/-} mice) the authors demonstrate that the IFN-I-independent downstream effects of IRF-5

activation are greatly contributing to the pathogenesis of the disease and that blockade of IRF-5 (via conditional *Irf5* deletion or using a small molecule inhibitor) suppresses disease progression. It is further suggested that the implication of IRF-5 in oxidative phosphorylation could contribute to disease development.

This study is well conducted and clearly presented. It builds on a previous work published in *Immunity* that reported, among others, that monoallelic deletion of *Irf5* alleviated the development of SLE-like symptoms in *Lyn*^{-/-} mice. Although the clinical relevance and the therapeutic potential of the current results are evident, the manuscript seems unfinished and somehow incongruent. For example, the role of IRF-5 in regulating oxphos is suggested but not explored. Expression of IRF-5 in monocytes is reported as important but it is also not explored; this could easily be done generating cell-specific KO mice using *Irf5* flox/flox mice. Moreover, there is no attempt to explain how do all these components (monocytes, antibodies, oxphos) tie up together.

1) Methods:

Western blots: it would be better to show original western blots; bands seem too sharp right now.

qPCR data: the method of calculating mRNA levels should be better explained. Are these fold inductions or $\Delta\Delta CT$? Healthy control dots should also be shown.

IRF-5 translocation: how many cells were counted per cell type to determine the percentage of nuclear translocation?

Glomeruli: Counting 5 glomeruli per mouse is too little and could lead to a biased analysis; how many microscopic fields were counted? The number of glomeruli should be increased. Fig. 2f what is relative MFI?

2) Fig. 2g-h: the fact that the inflammatory and the IFN- γ responses were increased in *Lyn*^{-/-}*Irf5*^{-/-} compared to *Lyn*^{-/-} *Ifnar*^{-/-} mice was unexpected. Does this mean that IFN-I reduces inflammation during SLE?

3) Results on the oxphos genes should be validated. Is this important for SLE pathogenesis? Where does this regulation occur? Monocytes or other cells? This could easily be shown.

4) Fig. 4 A and E: it would be better to compare TAM-treated vs TAM-untreated mice in the same graph.

Response to the reviewers' comments

We thank all the reviewers for their careful review of our manuscript and valuable comments. As per their remarks, we have extensively performed additional experiments and revised the manuscript accordingly.

The main additions/revisions are as follows:

(1) To investigate the cell type-specific mechanisms differentially regulated by IRF5 and type I IFNs in SLE mice, we performed the following experiments:

- We analyzed the activation of immune cells from *Lyn*-deficient mice with mono- or biallelic deletion of *Ifnar1* or *Irf5* (Supplementary Fig. 2a–d), or with conditional deletion of *Irf5* (Supplementary Fig. 3c,d). We showed that the activation of cDC2s and CD4⁺ T cells in *Lyn*-deficient mice was ameliorated by IRF5 deficiency more effectively than by IFNAR1 deficiency (described on pages 6–7). Furthermore, immune cell activation was suppressed by IRF5 deficiency after disease onset (page 9).
- In immune cells from the same mice as above, we measured the mitochondrial membrane potential, which reflects the process of OXPHOS, and is therefore, a key indicator of mitochondrial activity (Supplementary Fig. 2j and 3e). We showed that IRF5-dependent mitochondrial dysregulation occurred in Ly6C⁺ monocytes and pDCs, possibly at an early stage of disease development (pages 7–9).

Taking the above results together, we have added a discussion regarding how IRF5 and OXPHOS fit into the scheme of SLE pathogenesis (page 14).

(2) We evaluated the effects of the IRF5 inhibitor alone, without remission induction, in SLE model mice (Supplementary Fig. 6a–k) and showed that the IRF5 inhibitor prevented the progression of SLE-like symptoms both before and after disease onset (page 11).

(3) We analyzed the RNA-seq data of *Lyn*-deficient mice with monoallelic *Irf5* deletion (Fig. 2j,k and Supplementary Fig. 2g,h) and showed that OXPHOS gene expression was markedly lower in *Lyn*^{-/-}*Irf5*^{+/-} mice than in *Lyn*^{-/-}*Ifnar1*^{-/-} mice (page 7).

(4) To confirm the specificity of our newly generated monoclonal antibody against phospho-IRF5, we showed that the IKKβ-induced IRF5 phosphorylation detected by the antibody disappeared when Serine 446 of IRF5 was substituted by alanine (Supplementary Fig. 1m).

(5) We measured the blood urine nitrogen (BUN) levels in serum samples and obtained results consistent with the proteinuria measurements (Supplementary Fig. 6g,l).

(6) In the H&E image analysis, we increased the number of glomeruli from five to ten per mouse,

referring to a previously described protocol (Xu et al, *Methods Mol Biol* 1134, 103-130, 2014). We updated the graphs in Figs 2e, 3h, and 6g using the data from ten glomeruli per mouse.

- (7) The text and figures have been revised in accordance with the comments and suggestions of the reviewers. The new sentences have been indicated using red font and the new figures are marked by red rectangles.

We believe that these revisions have significantly enhanced the quality of our manuscript, and we are very grateful to the reviewers for providing us with the opportunity to perform these experiments. Please find our point-by-point responses below.

Reviewer #1

Immunosuppressive agents widely used for treating SLE globally inhibit the immune system which predispose patients to severe infection and morbidity. Although significant progress made towards developing targeted therapies for SLE, and several novel therapeutics have reached Phase 2 and 3 clinical trials, to date, only one drug with modest efficacy (Belimumab) has been approved by the FDA. Much anticipated type I IFN blocking therapy has proven some success, but not satisfactory, thus necessitating the development of an improved therapy for SLE. A major roadblock to the development of therapeutics that are specific, effective and have fewer or no side effects is our poor understanding of the cell-intrinsic mechanisms of innate and adaptive immune cells that promote SLE pathogenesis. Delineation of novel critical signaling mechanisms involved in SLE is required to identify therapeutically targetable factors for SLE treatment. With that in mind the current manuscript by Ban et al. determined the potential for targeting IRF5 as an improved therapeutic target for SLE. Studies carried out by this group using both SLE patient samples and mouse models of SLE indicated IRF5 to be a superior therapeutic target for SLE than blocking type IFN signaling. The authors demonstrated that blocking Irf5 could be an effective therapeutic target during the onset or the remission phase of disease.

Major Comments:

(1) Although the authors have convincingly demonstrated the potential for IRF5 to be a better therapeutic target for SLE, the cell-type specific mechanisms by which IRF5 and type I IFN signaling may differentially regulate innate and adaptive immune responses to promote SLE autoimmunity was not investigated. The discovery that IRF5 regulates pathways other than just driving type IFN response is significant and novel, and therefore, delineating mechanisms

differentially regulated by IRF5 and type I IFNs (IFN-I) in a cell-type specific manner will fill gap in our knowledge of IRF5 and IFN-I functions.

Response 1-1: We appreciate the reviewer's important comments. To address this issue, we analyzed the cell surface markers of innate and adaptive immune cell activation, that is, CD80 on cDC1s, cDC2s, and pDCs, and CD69 on CD4⁺ T cells and CD8⁺ T cells in WT, *Lyn*^{-/-}, *Lyn*^{-/-}*Ifnar1*^{+/-}, *Lyn*^{-/-}*Ifnar1*^{-/-}, *Lyn*^{-/-}*Irf5*^{+/-}, and *Lyn*^{-/-} *Irf5*^{-/-} mice. We found that CD80 on cDC2s and CD69 on CD4⁺ T cells were upregulated in *Lyn*^{-/-} mice compared to WT mice, and this upregulation was suppressed by IRF5 deficiency more effectively than by IFNAR1 deficiency (Supplementary Fig. 2c,d). We also analyzed the cell type-specific mitochondrial activity (see Response 1-6 for a discussion on how IRF5 and IFN-I fit into the scheme of SLE pathogenesis).

*(2) Through the analysis of transcriptomes, the authors have identified OXPHOS metabolic genes to be differentially regulated in peripheral blood cells between healthy control and SLE patients, and *Lyn*^{-/-}*Ifnar1*^{-/-} and *Lyn*^{-/-}*Irf5*^{-/-} mice. However, no experiments were performed to validate these genes at the protein levels in a cell-type specific manner.*

Response 1-2: Since the GSEA results represent changes in more than 30 OXPHOS genes, we investigated the activity of mitochondria as a protein-level validation of these genes. We measured the mitochondrial membrane potential, which reflects the process of OXPHOS, and is therefore, a key indicator of mitochondrial activity in various cell types. We used tetramethylrhodamine methyl ester (TMRM) as a probe for the mitochondrial membrane potential. Our new data showed that the TMRM signals in Ly6C⁻ monocytes and pDCs from *Lyn*^{-/-} mice were higher than those from WT mice, and these high TMRM signals were suppressed by IRF5 deficiency but not by IFNAR1 deficiency (Supplementary Fig. 2j). These results indicate that IRF5 is indeed involved in the regulation of mitochondrial activity in certain cell types during SLE.

*(3) In Figure 3, the authors used tamoxifen inducible *Irf5* conditional knockout system to demonstrate at what stage of targeting IRF5 may have therapeutic benefit. Although the authors focused on autoantibody and type I IFN responses, and kidney pathology, they have missed opportunities to determine the innate and adaptive cellular immune responses to see what cell populations might be targeted.*

Response 1-3: We analyzed the cell-surface markers of innate and adaptive immune cell activation (CD80 and CD69 on DCs and T cells, respectively). We found that CD80 and CD69, especially, on cDC2s and CD4⁺ T cells, were upregulated in *Lyn*^{-/-} mice. The activation of cDC2s and CD4⁺ T cells was prevented by IRF5 deficiency more effectively than by IFNAR1 deficiency (Supplementary Fig. 2c,d). Furthermore, the activation of these cell types was suppressed by

conditional IRF5 deficiency after disease onset (Supplementary Fig. 3c,d).

(4) They have identified a novel prototypical inhibitor for IRF5 which can inhibit IRF5 activation, type I IFN induction and ameliorate kidney disease. Again, no cellular responses were investigated to determine the cell type specific effects of this inhibitor.

Response 1-4: We analyzed the cell-surface markers of innate and adaptive immune cell activation (CD80 and CD69) and mitochondrial activity (TMRM) in NZB/W F1 mice. Unexpectedly, increases in CD80 on cDC2s, CD69 on CD4⁺ T cells, and the TMRM signals in Ly6C⁻ monocytes and pDCs in DMSO-treated NZB/W F1 mice were not evident compared to WT C57BL/6 mice and YE6144-treated NZB/W F1 mice (Supplementary Fig. 6i-k). This is somehow a contrast to the fact that SLE symptoms, such as the presence of anti-dsDNA IgG, proteinuria, and high BUN levels were evident in NZB/W F1 mice and all of them were suppressed by the IRF5 inhibitor (Supplementary Fig. 6d,f,g). We infer that the mechanism of the disease in NZB/W F1 mice may be somewhat distinct from that in the Lyn-deficient mouse model. Since the upregulation of OXPHOS genes was clearly observed in human SLE patients, Lyn-deficient mice appear to be the more relevant mouse model in the context of OXPHOS in SLE. These results are described on page 11.

(5) It is not clear whether the novel IRF5 inhibitor the authors have identified ameliorates disease by dampening the OXPHOS metabolic activity in B cells or myeloid cells (such as monocytes or DCs)

Response 1-5: For the reasons mentioned above, we analyzed the mitochondrial activity in *Lyn*^{-/-} mice. As a premise of testing the inhibitor, we first checked whether *Irf5* conditional knockout suppressed TMRM signals after disease onset. In contrast to the straight *Irf5* knockout (*Lyn*^{-/-}*Irf5*^{-/-}), the effect of conditional *Irf5* ablation after disease onset in *Lyn*^{-/-} mice on the aberrant TMRM signals in Ly6C⁻ monocytes and pDCs appeared to be limited (Supplementary Fig. 3e). We therefore did not test YE6144 on mitochondrial activity in *Lyn*^{-/-} mice. These results imply that IRF5 is involved in a process that causes abnormal mitochondrial activity at an early stage of SLE pathogenesis. Once SLE-like disease fully develops, the regulation of mitochondrial activity becomes IRF5-independent, stressing the importance of induction therapy prior to IRF5 inhibition. We have described this on page 9.

In conclusion, although authors have identified IRF5 to be a superior therapeutic target for SLE than type I IFN blocking therapy, the manuscript lacks cell-intrinsic mechanisms through which IRF5 targeting may have better therapeutic effects than type I IFN blocking approach.

Understanding the differential cell intrinsic mechanisms of IRF5 and type I IFN signaling will fill gap in our knowledge about the role of IRF5 beyond just driving type I IFN response.

Response 1-6: In the original version of our manuscript, we showed that IRF5 is involved in the induction of the expression of both ISG and OXPHOS genes in SLE pathogenesis. Based on the reviewer's important suggestions, we investigated the cell type-specific roles of IRF5, and found that IRF5, but not IFNAR1, is required for the activation of cDC2s and CD4⁺ T cells in the Lyn-deficient SLE model. In addition, we now show that IRF5 may be uniquely involved in the mitochondrial dysregulation of Ly6C⁻ monocytes and pDCs, possibly at an early stage of disease development. Mitochondrial hyperpolarization has been shown to induce the production of reactive oxygen species (ROS) (Pearce et al, *Nat Rev Immunol* 15, 18-29, 2015). Given that dysfunctional mitochondria with excessive ROS levels are prone to extrude mitochondrial DNA into the cytoplasm to stimulate the TLR9, cGAS-STING, or inflammasome pathway (Riley and Tait, *EMBO Rep* 21, e49799, 2020), we envisage that IRF5-mediated mitochondrial dysfunction in these innate immune cells may promote SLE pathogenesis (described on page 14). Nevertheless, we agree that there will be many interesting issues to be pursued in the future, and we would like to determine the direct target genes of IRF5 in the relevant cell types, such as monocytes, pDCs, and CD5⁻CD163⁺CD14⁺ cells as a future research subject.

Reviewer #2

This paper expands previous knowledge by presenting a prototypical small-molecule compound that limits IRF5 activation to treat lupus in mice and hopefully in humans. In addition they report that the activation of IRF5 occurs in people with active and inactive disease and that treatment does not matter.

1. The efficacy of the IRF5 depends on the presence of a proteasome inhibitor and this should be stressed in the abstract.

Response 2-1: We thank the reviewer for raising this important point. Based on a previous study (Neubert et al, *Nat Med* 14, 748-755, 2008), we utilized the proteasome inhibitor bortezomib (BZ) to quickly induce remission in the mouse SLE models; thus, any methods that could induce remission would be useful. Because we already stated in the abstract that "IRF5 is effective for maintenance of remission in mice", we would like to retain this section as is, rather than emphasizing the specific method of induction therapy. This is also because we are not sure whether BZ is the right way to induce remission in human SLE. We would also like to mention that the conditional ablation of *Irf5* without using BZ suppressed the progression of SLE-like disease in

the *Lyn*-deficient mouse SLE model (Fig. 3a-i) and the IRF5 inhibitor alone also suppressed disease progression in NZB/W F1 mice (new data; Supplementary Fig. 6a-h).

2. *Fig. 2, Does the monoallelic Irf5 mouse have any Irf5? If so how much. How much should be present to reverse the beneficial effect?*

Response 2-2: In our previous study (Ban et al, *Immunity* 45, 319-332, 2016), we showed that deletion of a single *Irf5* allele reduced the IRF5 protein levels by 50%. Thus, we infer that the threshold of the protein level or activity of IRF5 that reverses the beneficial effect is 50%–100%.

3. *The monoallelic IRF5 animals should be included in Fig 2g through 2j.*

Response 2-3: We analyzed the RNA-seq data of *Lyn*-deficient mice with monoallelic *Irf5* deletion (Fig. 2j,k and Supplementary Fig. 2g,h) and showed that OXPHOS gene expression was significantly lower in *Lyn*^{-/-}*Irf5*^{+/-} mice than in *Lyn*^{-/-}*Ifnar1*^{-/-} mice (described on page 7).

4. *Figure 3, would KO of IRF5 in B cells accomplish the same effect? The Lyn-/- mouse develops lupus on a mechanism that starts directly from B cells (removal of the inhibitory effect of the kinase Lyn) and it is probable that IRF5 has a major effect on B cells at least in this model.*

Response 2-4: It has been reported that the B cell-specific loss of *Lyn* exhibits SLE-like symptoms and the additional deletion of *Myd88* in B cells ameliorates the disease (Lamagna et al, *J Immunol* 192: 919-928, 2014). Interestingly, however, similar results were also obtained following the DC-specific loss of *Lyn* and *Myd88* (Lamagna et al, *Proc Natl Acad Sci USA* 110: E3311-3320, 2013). Since IRF5 is activated in the TLR-MyD88 pathway, we infer that the KO of *Irf5* either in B cells or DCs would exert the same effects as those obtained following the aforementioned *Myd88* KO. IRF5 functions in multiple cell types involved in SLE pathogenesis, such as cDCs, pDCs, follicular DCs, monocytes, and B cells. Although this is not within the scope of this study, it will be interesting to determine whether therapeutic effects can be achieved by targeting IRF5 in a certain cell type. Nevertheless, we believe that showing the effects of targeting IRF5 in the entire body, as examined in this study, would be the first important step for the practical application of our concept.

5. *In Fig 5, dose curves for the drug should be provided.*

Response 2-5: As per the reviewer's request, we have provided the dose curves (Fig. 5f and Supplementary Fig. 5b).

6. *Fig 6. What is the effect of the IRF5 inhibitor alone (before and after the initiation of the disease).*

Response 2-6: We thank the reviewer for this important question. We analyzed NZB/W F1 mice treated with YE6144 alone before and after the initiation of the disease. The results indicated that the IRF5 inhibitor alone prevented the progression of the disease (Supplementary Fig. 6a–k, page 11).

7. *Would a proteasome inhibitor enhance the effect of IFN blockade? If that would be true then the authors can claim a better effect for their IRF5 inhibitor.*

Response 2-7: We used a proteasome inhibitor to quickly deplete long-lived plasma cells and induce remission in the mouse SLE models (see Response 2-1 above). Although we do not have data suggesting that a proteasome inhibitor directly enhances the suppression of type I IFNs and/or IRF5, this is an interesting idea for future investigations.

Reviewer #3

This manuscript clearly shows a pivotal role of persistent IRF5 hyperactivation in the initiation and perpetuation of SLE manifestations by in vitro studies of SLE patients and by using two lupus models of Lyn deficient mice and NZBxNZW F1 mice. Results are striking that partial inhibition of IRF5 exhibited superior efficacy than full inhibition of type I interferon signaling in suppressing murine lupus development, which has important clinical implications and highlights IRF5 as an excellent therapeutic target for a number of autoimmune diseases with a strong component of type I interferon mediated disease manifestations. The authors further developed a prototype small-molecule inhibitor of IRF5 activation and showed specific inhibition of nuclear transport and activation of IRF5. Treating established disease with either the inhibitor or conditional ablation of IRF5 could result in sustained remission of both murine models of lupus, which is impressive data to this reviewer.

The strength of the manuscript is its innovation and clinical significance that has direct implications to disease management of patients affected with SLE. The authors offered a mechanistic explanation on consequences of IRF5 hyperactivation in peripheral blood cells relating to expression levels of OXPHOS genes; however, it is not clear how this new information fits into the overall scheme of lupus pathogenesis at present.

Response 3-1: We thank the reviewer for this important comment. To address this issue, we measured the mitochondrial membrane potential, which reflects the process of OXPHOS, and is therefore, a key indicator of the mitochondrial activity in various cell types. Our new data showed

that the mitochondrial activity in Ly6C⁻ monocytes and pDCs from *Lyn*^{-/-} mice was higher than that in WT mice, and this upregulation was suppressed by IRF5 deficiency but not by IFNAR1 deficiency (Supplementary Fig. 2j). It has been shown that mitochondrial hyperpolarization, that is, an increase in the mitochondrial membrane potential, induces reactive oxygen species (ROS) production (Pearce et al, *Nat Rev Immunol* 15, 18-29, 2015). Mitochondrial ROS cause the release of mitochondrial DNA, which activates the TLR9, cGAS-STING, or inflammasome pathway (Riley and Tait, *EMBO Rep* 21, e49799, 2020). Taken together, we envisage that IRF5-mediated mitochondrial dysfunction might promote the vicious cycle of SLE, in which immune complexes containing autoantibodies and nucleic acids stimulate multiple types of cells and induce cytokines that further activate immune cells (described on page 14). We would like to determine the direct target genes of IRF5 in the relevant cell types, such as monocytes, pDCs, and CD5⁻CD163⁺CD14⁺ cells in the future.

An important reagent used in this study is a monoclonal antibody to phosphorylated IRF5 that appeared to be made by the authors who immunized rabbits with a peptide containing phosphorylated S446 of human IRF5 isoform B. It would be important for the authors to demonstrate specificity of this reagent in the supplemental materials, and to clarify how this S446 residue relates to the Serine 462 residue commonly described in the literature that its phosphorylation results in IRF5 dimerization and nuclear translocation.

Response 3-2: We agree with the reviewer. We confirmed the specificity of our newly generated monoclonal antibody against phosphor-IRF5 by showing that the IKK β -induced IRF5 phosphorylation detected by the antibody disappeared when S446 was substituted with alanine (Supplementary Fig. 1m). The S462 of human IRF5 isoform D is equivalent to the S446 of human IRF5 isoform B (Ren et al, *Proc Natl Acad Sci USA* 111, 17438-17443, 2014). In the revised manuscript, we also showed that our antibody detected the phosphorylation of IRF5 isoform D (Supplementary Fig. 1m; described in the Methods section).

Major findings of this manuscript corroborate with two recent publications that one links endosomal TLRs to the recruitment and activation of IRF5 independent of NF- κ B or MAPK via SLC15A4 and TASL (PMID:32433612) and the other on the development of cell-penetrating peptide to inhibit IRF5 activation and function in disease (PMID: 32440537), and the inclusion of these two could strengthen the Discussion of this manuscript.

Response 3-3: We appreciate the reviewer's suggestion. Per this comment, we have mentioned these two new papers (pages 10 and 15).

Additional specific comments are as follows:

1. The sex of the mice used should be stated. The authors showed spleen weights of the mice, which might be affected if the mice had different body weights, especially if both sexes were used. A common way to eliminate potential confounding is to present the spleen weight to body weight ratios.

Response 3-4: As pointed out by the reviewer, we have stated the sex of mice in the figure legends. We have also separately displayed the spleen weight according to the sex of the mice and confirmed that there was no remarkable gender-associated difference.

2. The authors measured proteinuria using Albustix, which is not very accurate. If urine samples are available, I would suggest to measure albumin creatinine ratio. Alternative, levels of blood urine nitrogen could be assessed in the serum samples.

Response 3-5: Although Albustix has been used in multiple studies (e.g., Kansal et al, *Sci Transl Med* 11, eaav1648, 2019; Singh et al, *J Exp Med* 183, 1613-1621, 1996; Ramos-Barrón et al, *Lupus* 16, 775-781, 2007), we agree with the reviewer's comment regarding the low accuracy of using Albustix. Therefore, we measured the blood urine nitrogen levels in serum samples and obtained consistent results (Supplementary Fig. 6g, I).

Reviewer #4

*The manuscript by Ban et al. investigates the activation status and function of the transcription factor IRF-5 during systemic lupus erythematosus (SLE) in SLE patients and in a murine model of SLE. Analysis of IRF-5 and interferon-stimulated gene expression revealed that both are activated not only in active SLE patients but also in patients in remission. Using a murine model for SLE (*Lyn*^{-/-} mice) the authors demonstrate that the IFN-I-independent downstream effects of IRF-5 activation are greatly contributing to the pathogenesis of the disease and that blockade of IRF-5 (via conditional *Irf5* deletion of using a small molecule inhibitor) suppresses disease progression. It is further suggested that the implication of IRF-5 in oxidative phosphorylation could contribute to disease development.*

*This study is well conducted and clearly presented. It builds on a previous work published in Immunity that reported, among others, that monoallelic deletion of *Irf5* alleviated the development of SLE-like symptoms in *Lyn*^{-/-} mice. Although the clinical relevance and the therapeutic potential of the current results are evident, the manuscript seems unfinished and somehow incongruent. For*

example, the role of IRF-5 in regulating oxphos is suggested but not explored.

Response 4-1: We thank the reviewer for raising this important point. We conducted additional experiments to analyze the cell type-specific mitochondrial activity (see Response 4-9 for a discussion on how IRF5 and OXPHOS fit into the scheme of SLE pathogenesis).

Expression of IRF-5 in monocytes is reported as important but it is also not explored; this could easily be done generating cell-specific KO mice using Irf5 flox/flox mice.

Response 4-2: In this study, we showed that IRF5 is activated in monocytes, pDCs, and cDCs (Fig. 1b and Supplementary Fig. 1e). Thus, IRF5 functions in multiple cell types involved in SLE pathogenesis, for example, cDCs, pDCs, follicular DCs, monocytes, and B cells, which is one of the reasons why IRF5 is a strong candidate as a target molecule. We believe that showing the effects of targeting IRF5 in the entire body, as examined in this study, would be the first important step for the practical application of the concept. Nevertheless, determining whether therapeutic effects can be achieved by targeting IRF5 in certain cell types is indeed an interesting issue that warrants future investigations.

Moreover, there is no attempt to explain how do all these components (monocytes, antibodies, oxphos) tie up together.

Response 4-3: We appreciate the reviewer's important comments. Per this suggestion, in the revised manuscript, we have discussed how these components tie up together (see Response 4-9).

1) Methods:

Western blots: it would be better to show original western blots; bands seem too sharp right now.

Response 4-4: As explained in the figure legends and Methods, we analyzed the protein expression not by the conventional western blotting method, but by a capillary-based immunoassay (Wes system; ProteinSimple). Proteins were separated by size in the capillary, immobilized to the capillary wall, and then immunoassayed. The resulting chemiluminescent signals were detected, quantitated, and virtually displayed, as seen in the traditional gel-blot image (band). Thus, the sharp bands were due to the sharp peaks obtained by the capillary system. In the revised manuscript, we have provided the original images of the capillary-based immunoassay at the end of the Supplementary information (page 65).

qPCR data: the method of calculating mRNA levels should be better explained. Are these fold inductions or $\Delta\Delta CT$? Healthy control dots should also be shown.

Response 4-5: As described in the Methods section, we used the $\Delta\Delta$ CT method to normalize the expression levels of the target genes to those of the housekeeping genes indicated in each graph. The healthy control dots are shown in Fig. 1c.

IRF-5 translocation: how many cells were counted per cell type to determine the percentage of nuclear translocation?

Response 4-6: We counted 75–100 cells per cell type for each individual, as explained in the Methods section.

Glomeruli: Counting 5 glomeruli per mouse is too little and could lead to a biased analysis; how many microscopic fields were counted? The number of glomeruli should be increased. Fig. 2f what is relative MFI?

Response 4-7: In the H&E image analysis, we counted nuclei per glomerulus for five glomeruli from three to five microscopic fields for each mouse. As suggested by the reviewer, we increased the number of glomeruli from five to ten per mouse, referring to a previously described protocol (Xu et al, *Methods Mol Biol* 1134, 103-130, 2014), and confirmed the results. Accordingly, we have updated the graphs in Figs 2e, 3h, and 6 g. In the IgG immunofluorescence image analysis, we obtained one glomerulus from one microscopic field for five glomeruli per mouse. The MFI of each glomerulus was calculated and the MFI from five glomeruli were averaged. The relative MFI is the average MFI normalized to the average MFI of the control (wild-type C57BL/6). Since the fluorescent signal faded easily, it was difficult to increase the number of glomeruli. However, as observed in the case of the H&E image analysis, we expect that five glomeruli are sufficient to evaluate IgG deposition. We have now fully described how we calculated the relative MFI in the Methods section (page 22).

*2) Fig. 2g-h: the fact that the inflammatory and the IFN- γ responses were increased in *Lyn*^{-/-}*Irf5*^{-/-} compared to *Lyn*^{-/-} *Ifnar*^{-/-} mice was unexpected. Does this mean that IFN-I reduces inflammation during SLE?*

Response 4-8: We infer that the reason why the inflammatory and IFN- γ responses were more strongly suppressed by IFNAR1 deficiency than by IRF5 deficiency in *Lyn*^{-/-} mice is that many of the genes listed in inflammatory responses and IFN- γ responses are also listed as IFN- α response genes (now indicated in Supplementary Fig. 2g), and type-I IFNs indeed promote SLE pathogenesis. As shown in Supplementary Fig 2h, the expression levels of IFN- α response genes in *Lyn*^{-/-}*Ifnar1*^{-/-} mice were lower than those in *Lyn*^{-/-}*Irf5*^{-/-} mice and WT mice. This is because IFNAR1 deficiency, but not IRF5 deficiency, causes the complete loss of type I IFN signaling.

However, although IFNAR1 deficiency reduces all three (inflammatory, IFN- γ , and IFN- α) responses more strongly than IRF5 deficiency, reducing IRF5 expression or activity was superior to blocking type I IFN signaling in suppressing SLE pathogenesis, indicating the advantage of using IRF5 as a therapeutic target.

3) *Results on the oxphos genes should be validated. Is this important for SLE pathogenesis? Where does this regulation occur? Monocytes or other cells? This could easily be shown.*

Response 4-9: Since the GSEA results represent changes in more than 30 OXPHOS genes, we investigated the activity of mitochondria to validate the results. We measured the mitochondrial membrane potential, which reflects the process of OXPHOS, and is therefore, a key indicator of the mitochondrial activity in various cell types. We used tetramethylrhodamine methyl ester (TMRM) as a probe for measuring the mitochondrial membrane potential. Our new data showed that the TMRM signals in Ly6C⁻ monocytes and pDCs from *Lyn*^{-/-} mice were higher than the TMRM signals in those from WT mice, and these high TMRM signals were suppressed by IRF5 deficiency but not by IFNAR1 deficiency. These results indicate that IRF5 is indeed involved in the regulation of mitochondrial activity in certain cell types during SLE pathogenesis (Supplementary Fig. 2j). It has been shown that mitochondrial hyperpolarization, that is, an increase in the mitochondrial membrane potential, induces reactive oxygen species (ROS) production (Pearce et al, *Nat Rev Immunol* 15, 18-29, 2015). These mitochondrial ROS cause the release of mitochondrial DNA, which activates the TLR9, cGAS-STING, or inflammasome pathways (Riley and Tait, *EMBO Rep* 21, e49799, 2020). Taken together, we envisage that IRF5-mediated mitochondrial dysfunction might promote the vicious cycle of SLE, in which immune complexes containing autoantibodies and nucleic acids stimulate multiple types of cells and induce cytokines that further activate immune cells (described on page 14). We would like to determine the direct target genes of IRF5 in the relevant cell types, such as monocytes, pDCs, and CD5⁻CD163⁺CD14⁺ cells in the future.

4) *Fig.4 A and E: it would be better to compare TAM-treated vs TAM-untreated mice in the same graph.*

Response 4-10: It has been reported that TAM treatment has a certain beneficial effect on SLE regardless of CreER (Stoeger et al, *Ann Rheum Dis* 62, 341-346, 2003). Therefore, we used *Irf5*^{flx/flx} (without *CreER*) mice treated with TAM as controls. We have added a brief description regarding this aspect in the revised manuscript (page 8).

REVIEWERS' COMMENTS

Reviewer #1 (Remarks to the Author):

Although the revised manuscript is improved and authors have addressed most comments, this reviewer still has the following comment.

One of the major and novel findings in this manuscript is how IRF5 regulates autoimmune responses, and disease onset and remission independent of type I interferon signaling. Through the analysis of RNAseq data in PBMCs from SLE patients and cells from lupus-prone Lyn^{-/-} mice in the original version of the manuscript the authors demonstrated that type I IFN independent IRF5 function was mediated by its role in regulating OXPHOS. To address the comment by this reviewer with regard to the cell-intrinsic role, in the revised version of the manuscript, the authors have analyzed the mitochondrial membrane potential of both innate and adaptive immune cells. They show that IRF5 mediated differential regulation of OXPHOS was restricted to Ly6C⁺ monocytes and pDCs, thus excluding such differential regulation in B cells.

IRF5 downstream of TLR signaling is thought to play an important role in B cell activation and autoimmune B cell responses. Therefore, it is surprising that the authors found no effects of IRF5 deficiency on B cell mitochondrial membrane potential. The authors should perform Seahorse analysis on B cells purified from IRF5 and IFN α R KO Lyn^{-/-} mice to measure OCR (Oxygen Consumption Rate) to definitively show that IRF5 mediated differential regulation of OXPHOS occurs only in innate cells in SLE-prone mice. This will make their argument stronger.

Reviewer #2 (Remarks to the Author):

The authors have done a great job in addressing all concerns.

Reviewer #3 (Remarks to the Author):

The revised manuscript has addressed successfully all comments raised previously.

Reviewer #4 (Remarks to the Author):

All my concerns have been adequately addressed.

Response to the reviewers' comments

We are very glad to learn the reviewers' positive comments on our revised manuscript. We again would like to state that the careful and high-quality review by the reviewers was essential for us to be able to improve the quality of our study. Please find our point-by-point responses to Reviewer #1 below.

Reviewer #1:

Although the revised manuscript is improved and authors have addressed most comments, this reviewer still has the following comment.

One of the major and novel findings in this manuscript is how IRF5 regulates autoimmune responses, and disease onset and remission independent of type I interferon signaling. Through the analysis of RNAseq data in PBMCs from SLE patients and cells from lupus-prone Lyn^{-/-} mice in the original version of the manuscript the authors demonstrated that type I IFN independent IRF5 function was mediated by its role in regulating OXPHOS. To address the comment by this reviewer with regard to the cell-intrinsic role, in the revised version of the manuscript, the authors have analyzed the mitochondrial membrane potential of both innate and adaptive immune cells. They show that IRF5 mediated differential regulation of OXPHOS was restricted to Ly6C⁺ monocytes and pDCs, thus excluding such differential regulation in B cells.

IRF5 downstream of TLR signaling is thought to play an important role in B cell activation and autoimmune B cell responses. Therefore, it is surprising that the authors found no effects of IRF5 deficiency on B cell mitochondrial membrane potential. The authors should perform seahorse analysis on B cells purified from IRF5 and IFN α R KO Lyn^{-/-} mice to measure OCR (Oxygen Consumption Rate) to definitively show that IRF5 mediated differential regulation of OXPHOS occurs only in innate cells in SLE-prone mice. This will make their argument stronger.

Response: We thank the reviewer for pointing out this important issue. Although we showed the involvement of IRF5 in mitochondrial dysregulation in Ly6C⁺ monocytes and pDCs, we do not exclude the possibility that IRF5 regulates OXPHOS in B cells. We speculate from the current study and previous reports that IRF5 may be activated in a certain B cell subpopulation that induces autoantibody production. We would also like to mention a technical issue. We have already tried the seahorse analysis of various cell populations from wild-type mice using a flux analyzer, but it was difficult to obtain constant results. This appeared to be due to the lengthy time required for

cell sorting, which affects the status of cells. Because multiple cell types from multiple genotypes need to be compared, it will be not possible to prepare the cells in a short time. Thus, we would like to add a description regarding this point in our manuscript: “Yet, it is still possible that IRF5 regulates OXPHOS genes in certain subpopulations within other cell types as well.” (page 7, the last line). It will be an interesting future subject to clarify the involvement of IRF5 and OXPHOS in B cells.

Reviewer #2:

The authors have done a great job in addressing all concerns.

Reviewer #3:

The revised manuscript has addressed successfully all comments raised previously.

Reviewer #4:

All my concerns have been adequately addressed.